The biochronology and palaeobiogeography of Baru (Crocodylia: Mekosuchinae) based on new specimens from the Northern Territory and Queensland, Australia

Yates Adam M. adamm.yates@nt.gov.au
Museum of Central Australia, Museum and Art Gallery of the Northern Territory , Alice Springs, NT , Australia
Young Mark
Electronic publication date: 2017 Jun 21
Publication date: 2017
Volume: 5
Electronic Location ID: e3458
Received 2016 Sep 22; Accepted 2017 May 23
Copyright: © 2017 Yates
Copyright year: 2017
Copyright holder: Yates
License: This is an open access article distributed under the terms of the Creative Commons Attribution License, which permits unrestricted use, distribution, reproduction and adaptation in any medium and for any purpose provided that it is properly attributed. For attribution, the original author(s), title, publication source (PeerJ) and either DOI or URL of the article must be cited.
License URL: https://creativecommons.org/licenses/by/4.0/

Keywords: Biochronology, Palaeobiogeography, Baru, Australia, Northern Territory, Queensland, Oligocene, Miocene, Mekosuchinae, Crocodylia

Funding: The author received no funding for this work.

==============================
New records of the Oligo–Miocene mekosuchine crocodylian, Baru, from Queensland and the Northern Territory are described. Baru wickeni and Baru darrowi are accepted as valid species in the genus and their diagnoses are revised. Both species are present in Queensland and the Northern Territory but are restricted in time, with B. wickeni known from the late Oligocene and B. darrowi from the middle Miocene. The broad geographic distributions and restricted time spans of these species indicate that this genus is useful for biochronology. The record of B. wickeni from the Pwerte Marnte Marnte Local Fauna in the Northern Territory establishes that the species inhabited the north-western margin of the Lake Eyre Basin (LEB) drainage system. More southerly Oligo–Miocene sites in the LEB contain only one crocodylian species, Australosuchus clarkae. The Pwerte Marnte Marnte occurrence of B. wickeni indicates that the separation of Baru and Australosuchus did not correspond with the boundaries of drainage basins and that palaeolatitude was a more likely segregating factor.

Introduction

The Cenozoic of Australia hosted an endemic radiation of crocodylians known as the Mekosuchinae (Willis, 1997a). Mekosuchine fossils are known from the Eocene through the Holocene (Willis, Molnar & Scanlon, 1993; Mead et al., 2002) but it was in the late Oligocene and early Miocene that the clade radiated to produce a variety of ecomorphological types including ziphodonts, short-faced dwarves and platyrostral (flat-snouted) aquatic generalists (Molnar, 1981; Willis & Molnar, 1991; Willis, 1993, 1997b). Baru Willis, Murray & Megirian, 1990 is a distinctive member of this Oligo–Miocene radiation of mekosuchines. Species of Baru are characterised by their large size (adult skull lengths of more than 500 mm), very large, non-ziphodont teeth and exceptionally robust, broad and altirostral (deep) snouts which suggests that they were aquatic predators specialising on large vertebrate prey (Willis, Murray & Megirian, 1990). The genus is known from the Oligocene and Miocene of Northern and Central Australia (Fig. 1A). It was first erected to encompass a single species, Baru darrowi Willis, Murray & Megirian, 1990, which was recorded from terrestrial carbonate deposits in both the Northern Territory and Queensland (Willis, Murray & Megirian, 1990). The holotype specimen consists of a large rostrum (Fig. 1B) from the Bullock Creek Local Fauna of the Northern Territory, while less complete specimens from Riversleigh Station (now Riversleigh World Heritage Area (WHA)) of Northwestern Queensland were designated as paratypes of the same species. The Riversleigh species were treated as the same species despite the absence of minutely crenulated dental carinae present in the holotype and proportional differences in the symphyseal region of the dentary (Willis, Murray & Megirian, 1990). At the time of description, it was recognised that the Riversleigh Baru-bearing sites were older than the Bullock Creek Local Fauna, but the age difference was not considered remarkable for a crocodylian species (Willis, Murray & Megirian, 1990).

Figure 1 Baru Willis, Murray & Megirian, 1990 and its distribution.

(A) Map of Australia showing Queensland and the Northern Territory and the sites where Baru has been found. Note that the occurrence at Alcoota represents a new species dating from the late Miocene and is not discussed in this paper. Occurrences relevant to this paper are marked with triangles. (B) NTM P86158, holotype rostrum of B. darrowi Willis, Murray & Megirian, 1990, the type species of the genus. Scale bar in B = 100 mm.

Later in the same decade, two additional species of Baru were described (Willis, 1997b) on the basis of new material from Riversleigh and further study of the previous specimens. Baru wickeni Willis, 1997b is a large species, similar in size to B. darrowi. Willis (1997b) specifically included the Riversleigh paratypes of B. darrowi in the hypodigm of this new species. Baru huberi Willis, 1997b is a much smaller form with a platyrostral snout that was found alongside B. wickeni and named in the same paper. New evidence from a related species in the Bullock Creek Local Fauna indicates that B. huberi belongs to a distinct lineage more closely related to other mekosuchines than to B. darrowi (M. Lee & A. Yates, 2017, in preparation). A revision of this species that will name a new genus is in preparation and the species will not be discussed any further here.

Our understanding of the stratigraphy and biochronology of the many vertebrate fossil sites in the Riversleigh WHA has greatly improved since Baru was first described. At the coarsest level, the Oligo–Miocene sites can be divided into four faunal zones labelled A, B, C and D (Travouillon et al., 2006). We also recognise that these zones roughly correspond with the late Oligocene, early Miocene, middle Miocene and early–late Miocene, respectively, based on biocorrelation with other mammal-bearing sites in Australian and direct radiometric dating of some Riversleigh sites (Travouillon et al., 2006; Woodhead et al., 2016). B. wickeni has only been identified in faunal zone A, mostly from D–D Site and White Hunter Site (Willis, 1997b). Faunal zone A includes extensive fluviatile and lacustrine calcarenites and micrites with a notable aquatic component to the preserved faunas, such as large crocodylians, turtles and lungfish (Archer et al., 1989; Creaser, 1997). Younger deposits from faunal zone B, C and D consist mostly of laterally restricted pond and cave deposits that formed on a palaeokarst landscape (Archer et al., 1989; Creaser, 1997; Arena et al., 2014). As a consequence, crocodylians are less common and are of smaller size than those in faunal zone A deposits. Nevertheless, a fauna dominated by aquatic taxa including a diverse assemblage of small crocodylians has been recovered from the faunal zone C Ringtail Site (Willis, 2001; Woodhead et al., 2016). This fauna includes a small maxilla and an even smaller dentary that Willis (2001) referred to Baru sp. Willis (2001) declined to identify these specimens to species level due to of their incompleteness. Specifically diagnostic Baru specimens from Riversleigh’s faunal zone C would be highly desirable because the zone shares many mammal species with the middle Miocene Bullock Creek Local Fauna (Travouillon et al., 2006; Megirian et al., 2010) and it would be important to extend that correlation to crocodylians as well. The question of the specific identity of these specimens is revisited in this paper.

As records currently stand both B. darrowi and B. wickeni are known only from their respective type localities, or, in the case of B. wickeni, a cluster of geographically proximate and temporally equivalent sites (Willis, 1997b). With only singular occurrences, little can be said about the geographic distribution or temporal range of these species. The difference in Baru species at the two localities may be due to the time difference between the two deposits (at least 10 million years) or the spatial separation (approximately 800 km) or a combination of both. Here, I report on new occurrences of both species that strongly suggest that they had broad geographic distributions across the region but were confined to non-overlapping ranges of time.

Systematic Palaeontology

Crocodylia Gmelin, 1789

Mekosuchinae (Balouet & Buffetaut, 1987)

Baru Willis, Murray & Megirian, 1990

Revised diagnosis: The original diagnosis of Baru was a short description that did not distinguish unique characters of Baru from those that are more broadly distributed among eusuchians. A new diagnosis is proposed here that is based upon characteristics that are interpreted as synapomorphies of the genus: deep, almost vertical profile of the premaxilla between the anterior alveolar margin and the external naris; four premaxillary teeth in all but the smallest post-hatching individuals due to the loss of the second premaxillay tooth (reversed in the undescribed Baru from Alcoota); a deep, semilunate convexity in the maxillary alveolar margin that encompasses maxillary alveoli 1–5; foramen for the posterior branch of the dorsal alveolar nerve opens dorsally near the dorsal margin of the maxilla (Fig. 2; unknown in B. wickeni); laterally projecting ridge bordering dorsal margin of ornamented area on the surangular (convergent in Mekosuchus).

Figure 2 Position and orientation of the maxillary foramen for the posterior branch of the maxillary nerve in various crocodylians.

(A) Baru sp. nov. from Alcoota, NTM P912, premaxillae, nasals and right maxilla in medial view. Scale bar = 50 mm. (B) Crocodylus johnstoni Krefft, NTM unregistered comparative specimen, right maxilla in posteromedial view. Note that the foramen for the posterior branch of the maxillary differs from that of Baru in being situated far from the dorsomedial margin of the maxilla and opening posteriorly. Scale bar = 20 mm. (C) Alligator mississippiensis Daudin, NTM R36716, skull in left ventrolateral view highlighting area enlarged in (D). Scale bar = 100 mm. (D) Alligator mississippiensis Daudin, NTM R36716, medial wall of right maxilla viewed through the suborbital fenestra. Note the similar position and orientation of the foramen for the posterior branch of the maxillary nerve to Crocodylus johnstoni. Scale bar = 20 mm. acr, aperture of the caviconchal recess; ec, ectopterygoid; eca, articular surface for the ectopterygoid; fdan, foramen for the main branch of the dorsal alveolar nerve; fpmn, foramen for the posterior branch of the maxillary nerve; la, articular surface for the lacrimal; m, maxilla; n, nasal; na, articular surface for the nasal; nc, narial canal; pal, palatine; pala, articular surface for the palatine; pm, premaxilla; sym, median symphyseal surface of the maxilla. Arrows indicate the orientation of the opening of the foramen for the posterior branch of the maxillary nerve.

Remarks: A compilation of all the material that has been referred to Baru in the literature (including the present paper) is given in Table 1.

Table 1 List of all specimens referred to the genus Baru Willis, Murray & Megirian, 1990 in the published literature, including the present paper.

Specimen	Description	Local Fauna/Site	Previous referrals	
Baru wickeni	
NTM P2914-15	Jugal fragment	Pwerte Marnte Marnte	No previous referral	
NTM P8681-14	Mandibular ramus	D Site, Riversleigh WHA*	B. darrowi (Willis, Murray & Megirian, 1990); B. wickeni (Willis, 1997a)	
NTM P8738-1	Skull and dentary fragments	D Site, Riversleigh WHA*	B. darrowi (Willis, Murray & Megirian, 1990); B. wickeni (Willis, 1997a)	
NTM P8778-2	Posterior mandible fragment	D Site, Riversleigh WHA*	B. darrowi (Willis, Murray & Megirian, 1990)	
NTM P8778-4	Skull fragment from temporal region	D Site, Riversleigh WHA*	B. darrowi (Willis, Murray & Megirian, 1990)	
NTM P8778-5	Palate and skull fragment	D Site, Riversleigh WHA*	B. darrowi (Willis, Murray & Megirian, 1990)	
NTM P902-4	Posterior skull table fragment	D-Site, Riversleigh WHA*	No previous referral	
NTM P911	Posterior mandibular fragment	D-Site, Riversleigh WHA*	No previous referral	
NTM P91171-1	Skull	300 BR, Riversleigh WHA	No previous referral	
QM F16822 (Holotype)	Rostral region of skull and associated postcrania	D Site, Riversleigh WHA	B. darrowi (Willis, Murray & Megirian, 1990); B. wickeni (Willis, 1997a)	
QM F16823	Jugal fragment	D Site, Riversleigh WHA*	B. wickeni (Willis, 1997a)	
QM F16824	Premaxillary fragments	D Site, Riversleigh WHA*	B. wickeni (Willis, 1997a)	
QM F16825	Dentary	D Site, Riversleigh WHA*	B. wickeni (Willis, 1997a)	
QM F16826	Dentary	D Site, Riversleigh WHA*	B. wickeni (Willis, 1997a)	
QM F31070	Dentary	D Site, Riversleigh WHA*	B. wickeni (Willis, 1997a)	
QM F31071	Posterior mandibular fragment	White Hunter, Riversleigh WHA	B. wickeni (Willis, 1997a)	
QM F31072	Posterior mandibular fragment	White Hunter, Riversleigh WHA	B. wickeni (Willis, 1997a)	
QM F31073	Dentary fragment	White Hunter, Riversleigh WHA	B. wickeni (Willis, 1997a)	
QM F31074	Skull fragments	Sticky Beak, Riversleigh WHA	B. wickeni (Willis, 1997a)	
QM F31075	Posterior skull fragment	White Hunter, Riversleigh WHA	Cranial Form 1 (Willis, 1997b); Ultrastenos willisi (Stein, Hand & Archer, 2016)	
Tentatively B. wickeni	
NTM P2815-various	Vertebral fragments and unassociated osteoderms	Pwerte Marnte Marnte	Crocodylia gen. et sp. indet. (Murray & Megirian, 2006)	
NTM P2815-18	Anterior tooth crown	Pwerte Marnte Marnte	Crocodylia gen. et sp. indet. (Murray & Megirian, 2006)	
NTM P2914-5	Anterior tooth crown	Pwerte Marnte Marnte	No previous referral	
NTM P2914-14	Posterior mandibular fragment	Pwerte Marnte Marnte	No previous referral	
NTM P2914-16 & 17	Angular fragments	Pwerte Marnte Marnte	No previous referral	
NTM P6372	Juvenile squamosal fragment	Pwerte Marnte Marnte	No previous referral	
NTM P6373	Posterior tooth crowns	Pwerte Marnte Marnte	No previous referral	
B. darrowi	
NTM P8695-8 (Holotype)	Incomplete skull	Bullock Creek	B. darrowi (Willis, Murray & Megirian, 1990)	
NTM P87103-12	Maxilla	Bullock Creek	No previous referral	
NTM P2786-7	Squamosal	Bullock Creek	No previous referral	
NTM P87115-15	Mandibular ramus	Bullock Creek	No previous referral	
NTM P8710-1	Posterior mandibular fragment	Bullock Creek	No previous referral	
QM F30319	Premaxilla and nasal	Ringtail, Riversleigh WHA	No previous referral	
QM F31185	Maxilla	Ringtail, Riversleigh WHA	Baru sp. indet. (Willis, 2001)	
QM F31013	Maxilla fragment	Ringtail, Riversleigh WHA	Baru sp. indet. (Willis, 2001)	
Baru sp. nov.	
NTM P5335	Incomplete skull	Alcoota	Baru sp. nov. (Yates & Pledge, 2017)	
NTM P912	Incomplete skull	Alcoota	No previous referral	
NTM P6515	Incomplete and disarticulated skull	Alcoota	No previous referral	
NTM P6319	Premaxilla	Alcoota	No previous referral	
Mekosuchinae gen. nov. huberi	
QM F31060 (Holotype)	Incomplete rostrum	White Hunter, Riversleigh WHA	B. huberi (Willis, 1997a)	
QM F31061	Rostral fragment	White Hunter, Riversleigh WHA	B. huberi (Willis, 1997a)	
QM F31062	Premaxilla	White Hunter, Riversleigh WHA	B. huberi (Willis, 1997a)	
QM F31063	Maxilla	White Hunter, Riversleigh WHA	B. huberi (Willis, 1997a)	
QM F31064	Maxillary fragment	White Hunter, Riversleigh WHA	B. huberi (Willis, 1997a)	
QM F31065	Maxillary fragment	White Hunter, Riversleigh WHA	B. huberi (Willis, 1997a)	
QM F31066	Maxillary fragment	White Hunter, Riversleigh WHA	B. huberi (Willis, 1997a)	
QM F31067	Dentary	White Hunter, Riversleigh WHA	B. huberi (Willis, 1997a)	
QM F31068	Dentary	White Hunter, Riversleigh WHA	B. huberi (Willis, 1997a)	
QM F31069	Anterior mandibular pair	White Hunter, Riversleigh WHA	B. huberi (Willis, 1997a)	
Mekosuchinae gen. et sp. nov. robust dentary form	
NTM P87103-11	Posterior skull fragment	Bullock Creek	B. darrowi (Willis, Murray & Megirian, 1990)	
Mekosuchinae gen. et sp. indet.	
QM F31004	Dentary	Ringtail, Riversleigh WHA	Baru sp. (Willis, 2001)	
Note:

* An asterix indicates that there is a suggestion of doubt over the provenance of these specimens due to a possible of mixing of samples from D Site and Don Camillo Site. However the balance of evidence is that these specimens are correctly attributed to D Site.

Baru wickeni Willis, 1997b

(Figs. 3B, 4–15, 16C–16E)

Figure 3 Comparison of the left jugals in the two named species of Baru Willis, Murray & Megirian, 1990.

(A) Baru darrowi Willis, Murray & Megirian, 1990, NTM P86158, fragment of the holotype skull including left maxilla, jugal and ectopterygoid. Red line indicates position of the section through jugal. (B) Baru darrowi, Willis, Murray & Megirian, 1990, outline of lateral surface of section through jugal (made by sectioning a cast) taken at the position indicated in (A). Black triangle indicates lateral jugal ridge. (C) Baru wickeni, Willis, 1997b, NTM P91171-1, portion of the left side of the skull, including the jugal. Red line indicates position of the section through jugal. (D) Baru wickeni, Willis, 1997b, outline of lateral surface of section through jugal (made by sectioning a cast) taken at the position indicated in (C). Note that the jugal ridge (indicated by the black triangle) is enlarged into what is here termed the ventrolateral flange.

Figure 4 Baru wickeni Willis, 1997b, NTM P2914-15, anterior fragment of the right jugal from Pwerte Marnte Marnte.

(A) Lateral view. (B) Medial view. (C) Anteromedial view. eca, articular surface for the attachment of the ectopterygoid; lf, lateral fossa; mjf, medial jugal foramen; om, orbital margin; vlf, ventrolateral flange; vm, ventral margin. Scale bar = 50 mm.

Figure 5 Mekosuchinae, tentatively Baru wickeni Willis, 1997b, NTM P6372, posterior fragment of the right squamosal of a small juvenile from Pwerte Marnte Marnte.

(A) Dorsal view. (B) Posterior view. (C) Ventral view. pa, articular surface for attachment of the parietal; plp, posterolateral process. Scale bar = 10 mm.

Figure 6 Baru, tentatively B. wickeni Willis, 1997b, NTM P2914-14, posterior region of left mandible from Pwerte Marnte Marnte.

(A) Lateral view. (B) Medial view. (C) Dorsal view. Scale bar = 50 mm.

Figure 7 Baru, tentatively B. wickeni Willis, 1997b, NTM P2914-14, interpretive drawings of posterior region of left mandible from Pwerte Marnte Marnte.

(A) Lateral view. (B) Medial view. (C) Dorsal view. Grey areas indicate patches of matrix and unassociated bone fragments. Hatched areas indicate broken bone surfaces. an, angular; ana, articular surface for the attachment of the angular; ar, articular; ara, articular surface for the attachment of the articular; dla, articular surface for attachment of dorsal lamina of articular; dlr, dorsolateral ridge of the surangular; mf, medial foramen for articular artery and alveolar nerve; rap, retroarticular process; sa, surangular; saa, articular surface for the attachment of the surangular. Scale bar = 50 mm.

Figure 8 Mekosuchinae, tentatively Baru wickeni Willis, 1997b, isolated teeth from Pwerte Marnte Marnte.

(A and B) NTM P2914-5, anterior tooth in (A) labial view and (B) posterior view. (C and D) NTM P6373, posterior tooth in (C) occlusal view and (D) labial view. c, carina. Scale bar = 10 mm.

Figure 9 Baru wickeni Willis, 1997b, NTM P91171-1, skull in dorsal view.

(A) Photograph. (B) Interpretive drawing. b, boss above root of fifth maxillary tooth; en, external naris; ex, exoccipital; f, frontal; itf, infratemporal fenestra; j, jugal; l, lacrimal; m, maxilla; n, nasal; om, bevelled orbital margin; p, parietal; pf, prefrontal; pm, premaxilla; po, postorbital; pob, postorbital bar; por, preorbital ridge; q, quadrate; qj, quadratojugal; so, supraoccipital; sq, squamosal; stfe, supratemporal fenestra; stfo, supratemporal fossa; vlf, ventrolateral flange of the jugal. Hatched areas represent broken bone surfaces, stippled areas represent adherent matrix, areas of light tone represent reconstructed areas and filler of plaster and glue; dark tone represents the palatal surfaces. Scale bar = 200 mm.

Figure 10 Baru wickeni Willis, 1997b, NTM P91171-1, skull in occipital view.

(A) Photograph. (B) Interpretive drawing. bo, bs, cqc, cranioquadrate canal; ec, ectopterygoid; ex, exoccipital; fm, foramen magnum; hf, hypoglossal foramen; j, jugal; lcf, lateral carotid foramen; lef, lateral Eustachian foramen; lvf, lateral vagus foramen; m, maxilla; mef, median Eustachian foramen; mvf, medial vagus foramen; oc, occipital condyle; p, parietal; pp, posterior process of the pterygoid; pt, pterygoid; pta, articulation surface for the pterygoid; q, quadrate; qa, articulation surface for the quadrate; qj, quadratojugal; so, supraoccipital; sq, squamosal; vmc, ventromedial crest of the quadrate. Hatched areas represent broken bone surfaces, stippled areas represent adherent matrix, areas of solid tone represent reconstructed areas and filler of plaster and glue. Scale bar = 200 mm.

Figure 11 Baru wickeni Willis, 1997b, NTM P91171-1, skull in ventral view.

(A) Photograph. (B) Interpretive drawing. 1–13, maxillary alveolar numbers; ap, alveolar process; as, adductor musculature scar; ch, choana; cqc, cranioquadrate canal; ec, ectopterygoid; ex, exoccipital; if, incisive foramen; j, jugal; m, maxilla; mf, maxillary foramen for palatine ramus of the trigeminal nerve; oc, occipital condyle; pal, palatine; pm, premaxilla; pp, posterior process of the pterygoid; pt, pterygoid; ptr, pterygoid ridge; q, quadrate; qa, articulation surface for the quadrate; qj, quadratojugal; rp, reception pit for dentary teeth; sof, suborbital fenestra; vmc, ventromedial crest of the quadrate. Note that the left alveolus 1 is present but invisible due to its position on the anterolateral surface of the maxilla. Hatched areas represent broken bone surfaces, stippled areas represent adherent matrix, areas of light tone represent reconstructed areas and filler of plaster and glue; dark tone represents the ventral surface of the dorsal skull roof. Scale bar = 200 mm.

Figure 12 Baru wickeni Willis, 1997b, NTM P91171-1, skull in left and right lateral views.

(A) Left lateral view. (B) Right lateral view. fest, anterior maxillary festoon; pp, posterior process of the pterygoid; vlf, ventrolateral flange of the jugal. Scale bar = 200 mm.

Figure 13 Baru wickeni Willis, 1997b, QM F16822, holotype, rostrum in dorsal view

Interpretive drawing of Fig. 12A in Willis (1997b) showing re-interpretation of sutural relationships. In particular note the posterior process of the maxilla inserting into the lacrimal. en, external naris; f, frontal; if, incisive foramen; j, jugal; l, lacrimal; m, maxilla; n, nasal; pf, prefrontal; pm, premaxilla; ppm, posterior process of the maxilla; vlf, ventrolateral flange of the jugal. Stippled areas represent remnants of adherent matrix, hatched areas represent broken bone surfaces, areas of solid tone represent bone surfaces below the dorsal skull roof. Scale bar = 50 mm.

Figure 14 Baru wickeni Willis, 1997b, NTM P8738-1, posterior end of right maxillary tooth row and surrounding bones in ventral view.

(A) Photograph. (B) Interpretive drawing of (A), supporting sutural interpretations in Fig. 11. 5–13, maxillary alveolus number; ec, ectopterygoid; j, jugal; m, maxilla; pml, posterior medial lamina of the maxilla; rp, reception pit for dentary teeth; vlf, ventrolateral flange of the jugal. Hatched areas represent broken bone surfaces, black areas represent deeply recessed, or internal bone surfaces. Scale bar = 50 mm.

Figure 15 Baru wickeni Willis, 1997b, NTM P902-4, skull table fragment from posterior margin in dorsal view.

(A) Photograph. (B) Interpretive line drawing. p, parietal; so, supraoccipital; sq, squamosal; stf, posterior margin of supratemporal fenestra. Scale bar = 20 mm.

Figure 16 Comparison of right squamosals of Baru Willis, Murray & Megirian, 1990.

(A and B) Baru darrowi Willis, Murray & Megirian, 1990, NTM P2786-7, from Bullock Creek in (A) ventral view and (B) dorsal view. (C and D) Mekosuchinae, tentatively Baru wickeni Willis, 1997a, NTM P6372, fragment from Pwerte Marnte Marnte in (C) ventral view and (D) dorsal view. (E) Baru wickeni, QM F31075, from White Hunter Site, Riversleigh World Heritage Area in dorsal view. Note the absence of a laterally protruding boss overhanging the lateral squamosal sulcus in the two Baru wickeni specimens. plb, posterolateral boss of the squamosal; plp, posterolateral process of the squamosal; stf, supratemporal fossa; vr, ventral rim of the lateral squamosal sulcus (highlighted with a blue line). Scale bars = 20 mm.

Revised diagnosis: Distinguished from B. darrowi by: nasals extend to margin of external naris; strongly developed preorbital ridge on lacrimal, flanked by lateral depression; deep ventrolateral ridge, forming pendent flange, extending from anterior end of jugal onto posterior end of maxilla (Fig. 3); posterolateral squamosal boss absent; sixth, seventh and eighth maxillary teeth separated by gaps wider than length of the seventh alveolus in adults; anterior process of palatines absent, palatine–maxilla suture linear to gently rounded; palatine–pterygoid suture level with posterior margin of suborbital fenestra; anterior tip of ectopterygoid deeply inserted into maxilla (for distance approximately equal to combined length of ninth and tenth maxillary alveoli);posterior pterygoid processes are elongate, finger-like projections in ventral view; dorsal posterior lobe of dentary symphyseal surface level with ventral lobe; splenial widely separated from symphyseal surface; all teeth with smooth carinae.

A median semilunate process on the posterior margin of the skull table is currently only known in B. wickeni among Baru species and so may diagnose this species. However the state of the occipital margin of B. darrowi is unknown, rendering the optimisation of this character ambiguous.

Type Locality: D Site, D Site Plateau, Riversleigh WHA, Queensland. Unnamed fluvio–lacustrine limestone, Riversleigh Local Fauna, late Oligocene. GPS coordinates for this site have been recorded with the QM. Note that Willis, Murray & Megirian (1990) erroneously give the locality for QM F16822, which would later be designated the holotype of B. wickeni, as Pancake Site, Riversleigh. Michael Archer, who led the expedition that recovered the specimen has confirmed that this specimen was found at D Site (M. Archer, 2017, personal communication).

New material: NTM P2914-14, posterior end of left mandible. NTM P2914-16 and 17, fragments of left angular. NTM P2914-15, fragment of right jugal. NTM P6372, fragment of right squamosal from small juvenile. NTM P2815-18, isolated anterior tooth crown. NTM P2914-5, isolated anterior tooth crown. NTM P6373, isolated posterior tooth crowns. NTM P2914-unnumbered, several osteoderms and incomplete vertebrae that are not described here. NTM P91171-1, a large almost complete cranium of an adult individual. NTM P902-4, fragment of skull table from posterior margin.

Locality and stratigraphic age of new material: All, except NTM P902-4 and NTM P91171-1, from railside borrow pit on the new Ghan railway line, approximately 40 km south of Alice Springs, Northern Territory. GPS coordinates for this site have been recorded with the NTM. Unnamed fluvial deposit of calcite-cemented sandstones and conglomerates, Pwerte Marnte Marnte Local Fauna, late Oligocene (Murray & Megirian, 2006). NTM P902-4 from D Site Riversleigh WHA, Queensland in an unnamed freshwater limestone unit in Riversleigh faunal zone A, late Oligocene (Archer et al., 2006). It has been suggested that the Riversleigh material from D Site that was prepared at the NTM was accidentally mixed with material from a separate Riversleigh site, Don Camillo Site. Unfortunately the person who carried out this preparation (D. Megirian) has since died and the NTM specimen register contains no mention of this possible mix-up. The only specimens recorded from Don Camillo site in the NTM (a few incomplete dromornithid hind limb bones) have a distinctly different style of preservation from the Baru material purported to be from D-Site. Furthermore the style of preservation of these Baru specimens matches that of incontrovertible D Site fossils in the QM, suggesting that the larger fossils (crocodylian skull pieces and dromornithid limb bones) were recognisable and could be linked to known limestone blocks prior to any mixing of samples. NTM P91171-1 from 300BR site, Riversleigh WHA, Queensland. From an unnamed freshwater limestone deposit, here argued to be part of Riversleigh faunal zone A, late Oligocene. GPS coordinates for these sites have been recorded with the QM.

Description of the Pwerte Marnte Marnte specimens: The jugal fragment (Fig. 4) is from the anterior end of the right jugal, immediately anterior to the postorbital bar. The fragment is 67 mm deep dorsoventrally, suggesting a large, deep-snouted crocodylian. The medial side bears an enlarged medial jugal foramen, with a maximum internal diameter of 6.7 mm, nestled against the anterior side of the internal buttress for the postorbital bar. The lateral external surface is ornamented with irregular pits and ridges. A broad shallow sulcus extends longitudinally under the orbital margin. This sulcus is separated from the ventral margin of the jugal by a well-developed longitudinal ridge. In more complete B. wickeni specimens (e.g. NTM P91171-1, P87381), this ridge begins on the jugal at about the level of the postorbital bars and extends anteriorly onto the maxilla for a distance of approximately 1 cm. As in other B. wickeni specimens (e.g. NTM P91171-1, P8738-1, P8778-4; QM F16822), the peak of this ridge in NTM P2914-15 is directed ventrolaterally while its ventral surface is slightly excavated, giving the ridge the form of a pendent flange. The squamosal fragment bears large rounded pits on its dorsal surface and lacks differentiated marginal ornament. The dorsolateral margin is slightly raised above the dorsal surface but not to the degree that could be described as a squamosal horn. In dorsal view it does not bulge laterally as the posterior end of the dorsolateral margin does in B. darrowi (NTM P2786-7; Figs. 15A and 15B) and an undescribed species of Baru from the late Miocene of Alcoota (NTM P6515).

The surangular of NTM P2914-14 is incomplete anteriorly and does not preserve any margins of the external mandibular fenestra. The anterior lateral surface is sculpted with elongate pits and ridges that become progressively deeper towards the level of the glenoid. Posterior to the glenoid the lateral surface of the surangular is smooth and unornamented. The dorsal edge of the ornamented area is thickened and forms a low, laterally projecting ridge that begins at a point presumably level with the posterior margin of the external mandibular fenestra and extends posterior to the level of the glenoid. This ridge is not as sharply defined as in other B. wickeni (e.g. NTM P911, P87105-1, P87115-15,) but this is a small difference of degree and could easily be the result of individual variation. Unfortunately the dorsal part of the surangular adjacent to the glenoid is badly damaged (Figs. 6C and 7C), so it is not possible to determine whether a dorsal pit was present as it is in B. wickeni and B. darrowi (e.g. NTM P911, P87115-15), or the height of the surangular extension up the posterior wall of the glenoid. Posterior to the glenoid the surangular tapers to form a thin splint that extends along the lateral surface of the retroarticular process, dorsal to the angular. The posterior end of the surangular is damaged and incomplete but facets on the angular and articular indicate that it would have extended close to the posterior end of the retroarticular process. Medially the surangular bears a large exposed sutural scar where part of the anterior articular has broken away (Figs. 6B and 7B). This scar reveals that the articular had a semilunate lamina that projected anteriorly on the medial surface of the surangular, immediately below the dorsal margin (the ‘crocodyline process’ of Aoki, 1992). Half of the opening for the lingual articular foramen is impressed into the surangular immediately ventral to the sutural scar, indicating that in life this foramen opened on the articular–surangular suture. Anteriorly and ventral to the lingual foramen is a small piece of the surangular that overlaps the medial surface of the angular indicating an oblique scarf joint was present between the two bones in this area. As this fragment extends to where the anterior tip of the articular would have lain, it indicates that the medial expression of the surangular–angular suture met the articular at its anterior tip. The posterior end of the angular is preserved on P2914-14 (Figs. 6 and 7). The lateral surface is flat and smooth from the glenoid region posteriorly to the end of the retroarticular process. Anterior of this level, the lateral surface ventral to the ornamented region of the surangular becomes undulate and pierced by a couple of large neurovascular foramina. Medially there is a trough-like sulcus extending along the anterior part of the angular fragment, adjacent to the ventral margin, into which the anterior process of the articular sits. A separate fragment from a more anterior part of the angular (NTM P2914-16), which may well-represent the same specimen as NTM P2914-14, shows that the lateral surface was ornamented with widely spaced deep pits. The ventral margin is broadly rounded. The articular is rather poorly preserved. It is crushed, broken and missing the medial edge of the retroarticular process, the entirety of the glenoid and the anterior tip of the anterior process. The anterior process is more elongate than in extant Crocodylus (compare Fig. 6C with Brochu, 1999; Fig. 33D) and the sutural line with the medial side of the surangular descends less steeply ventrally from the dorsal margin of the jaw. The dorsomedial surface of the anterior process forms a simple slightly concave surface that lies against the medial surface of the surangular. There is no longitudinal sulcus adjacent to the articular–surangular suture as is present in some mekosuchines such as ‘Baru’ huberi (QM F31072).

The isolated teeth (NTM P2914-5, P6373; Fig. 8) from the site have elliptical cross sections with a labiolingual width that varies between 85% and 75% of the anteroposterior length. The anterior and posterior edges, each bear a smooth carina (Fig. 8). Posterior teeth have low rounded outline in lingual view while anterior teeth are tall and conical with a slightly lingually curved tip.

Description of NTM P91171-1: The skull (Figs. 9–12) is 515 mm long when measured along the dorsal surface from the posterior margin of the skull table to the rostral tip of the premaxillae. It is triangular in dorsal profile (Fig. 9) while the cross section of the rostrum between the orbits and the premaxilla–maxilla suture is markedly angular and trapezoidal. The latter feature is a result of the flat, almost planar, dorsal surface of the rostrum that is offset from the lateral surfaces of the rostrum by a dorsolateral angle that is more sharply developed than in B. darrowi. It is has a deep rostrum with a height to width ratio of 0.45 measured at the level of the fifth maxillary tooth which is comparable to the ratio of 0.42 seen in B. darrowi (measurements in Table 2). In dorsal view, the rostrum is broad with a length to width ratio of 0.77 (using the dimensions described in Willis, Murray & Megirian (1990), Table 1) which is slightly greater than that of B. darrowi (0.72) and significantly greater than the holotype of B. wickeni (0.61). The transverse width of the premaxillae is almost half the total length of the rostrum (0.48) as it is in B. darrowi. The dorsal margin of the snout is flat in lateral view (Fig. 12), which contrasts with the concave margin of B. darrowi (Fig. 1B). The premaxillae are short and deep as in B. darrowi and present a vertical rostral surface. The premaxillary teeth form a D-shaped arcade that is wider than long. The left premaxilla preserves a complete row of four alveoli, and like the holotype of B. wickeni, the ancestral second tooth position has been lost (Willis, 1997b). The openings of the first three alveoli are circular with the third being the largest. The fourth alveolus has an oblique opening that breaches the anterior wall of the broad premaxilla–maxilla notch for the caniniform fourth dentary tooth. Dorsally, the external naris is trapezoidal with rounded corners and straight lateral margins that converge posteriorly (Fig. 9). The aperture is slightly wider than it is long with the widest point at the rostral end. Ventrally, the shape and extent of the incisive foramen cannot be fully determined due to missing bone, but it is evident from the short stretches of natural margin still preserved that the foramen was broad, elliptical and placed close to the rostral end of the premaxillae although it cannot be determined if it abutted the premaxillary tooth row or not. It may or may not have merged with the large circular pits placed anterolaterally to the foramen which received the first pair of dentary teeth in occlusion. These pits did not breach the dorsal surface of the premaxillae as they do in numerous crocodylians. There are no reception pits for the second and third dentary teeth developed on the premaxilla.

Table 2 Measurements (mm) of NTM P91171-1, Baru wickeni Willis, 1997b, skull.

Dimensions	Measurement	
Length of the skull (from posterior margin of the skull table to the rostral tip)	515	
Length of the Rostrum (from the anterior margins of the orbits to the rostral tip)	316	
Height of the premaxilla	66 (l), 72 (r)	
Height of the rostrum at the premaxilla-maxilla notch	52 (l), 53 (r)	
Height of the rostrum at the level of the fifth maxillary tooth	80 (l), 87 (r)	
Width of the external naris	56	
Length of the external naris	50	
Width of the premaxillae	152	
Length of the premaxillae from posterior tip of the posterodorsal process to the rostral tip	142 (l), 131 (r)	
Width of the rostrum at the premaxilla-maxilla notch	101	
Width of the rostrum at the level of the fifth maxillary tooth	192	
Minimum interorbital distance	50	
Length of the orbit	90 (l), 73 (r)	
Distance between the posteroventral corners of the orbits	169	
Width of the skull table between the posterior squamosal corners	232	
Length of the supratemporal fenestra	33 (l), 37 (r)	
Width of the supratemporal fenestra	23 (l), 30 (r)	
Minimum distance from the posterior margin of the supratemporal fenestra to the posterior margin of the skull table	44 (l), 46 (r)	
Minimum width of the dorsal parietal bridge between the supratemporal fossae	40	
Occipital height of the skull from the ventral ends of the posterior pterygoid processes to the dorsal margin of the skull table	187	
Distance between the lateral extremities of the posterior pterygoid processes	60	
Height of the posterior exposure of the pterygoids from the ventral margin of the basioccipital to the ventral ends of the posterior pterygoid processes	43	
Width of the occiput between the basal tuberae	56	
Height of the basioccipital from the floor of the foramen magnum to the dorsal margin of the median Eustachian opening	115	
Length of the suborbital fenestra	151 (l), 159 (r)	
Width of the suborbital fenestra	66 (l), 68 (r)	
Width of the pterygoid plate	230	
Width of the choana	33	
Length of the choana	27	
Note:

Measurements are of the bones as they are preserved. l, indicates that the measurement was taken on the left side; r, indicates the right.

In lateral view, the alveolar margin of the maxilla, immediately posterior to the premaxilla–maxilla notch, is produced ventrally into a large semicircular convexity (Fig. 12). This convexity bears maxillary teeth one to five. A lower more elongate posterior convexity encompasses maxillary teeth eight to twelve. A fossa is present on the lateral surface of the maxilla dorsal to the anterior end of the large anterior convexity. It is separated from the premaxilla–maxilla notch by an everted, low, rounded ridge forming the posterior border of the notch. Posteriorly this fossa is bordered by a rounded, laterally projecting boss that is developed over the base of the root of the large, caniniform, fifth maxillary tooth.

Posteriorly, the lateral surface of the maxilla bears a second larger fossa that lies ventral to, and accentuates the thick preorbital ridge on the lacrimal. The lacrimal–maxilla suture is partially missing on the left side and obscured by a patch of adherent matrix on the right, so that the posterior spur of the maxilla that inserts into lacrimal cannot be seen on either side. However, it is reconstructed as present (Fig. 9), because well-developed spurs are present in the holotype of B. wickeni (Fig. 13).

The right maxilla bears 12 alveoli, which is one less than other specimens of B. wickeni (e.g. NTM P8738-1; Fig. 15). The discrepancy is due to the absence of the usual first maxillary alveolus, so that the large canine peak is at the fourth, not fifth, alveolus. The first alveolus is present on the left alveolus of this specimen and if it were present on the right, there would be a total of 13 maxillary alveoli in accord with other B. wickeni. Loss of the first alveolus is an unusual condition and is not been reported in any other mekosuchine specimen. Despite maintaining the first alveolus, the left maxilla has even fewer alveoli than the right, bearing only 11. In this case, the small posteriormost alveolus (13th) is absent and the normal sixth alveolus absent due to a malformation of the tooth row between the alveolar convexities. These unusual and asymmetric losses of tooth positions may indicate the specimen is gerontic or simply malformed. An alveolar process (sensu Molnar, 1981) formed by interconnection and buttressing of the medial alveolar walls is present from the second to fifth maxillary alveoli (Fig. 11). No such wall is present posterior to the anterior convexity. The teeth between the anterior and posterior convexities are more crowded than in specimens of B. wickeni from smaller individuals (e.g. NTM P8738-1; Fig. 15) where there are two large gaps between the sixth, seventh and eighth teeth that are wider than the diameter of any of these alveoli. Nevertheless, a gap of 13 mm occurs between the original sixth and seventh alveoli (now the fifth and sixth, due to the loss of the first alveolus) which exceeds the diameter of the original seventh alveolus (10 mm). In contrast, the gap between these teeth in the holotype of B. darrowi is only one-third the diameter of the seventh alveolus.

Reception pits for the dentary teeth are absent in the anterior region of the maxilla where the alveolar process is present but are present between the levels of the sixth and tenth alveoli. On the right maxilla each reception pit is located medial to the maxillary tooth row and is level with the gaps between alveoli. The first pit is the deepest and has clearly defined margins, subsequent posterior reception pits become shallower and less clearly defined. However, on the left maxilla, the first reception pit is exceptionally large and deep and is located directly between the fifth and seventh alveoli, obliterating the sixth alveolus. This arrangement is an obvious malformation and is not present in any other specimen of Baru, or indeed the opposite side of the same specimen. A row of neurovascular foramina open along the base of the alveolar process, medial to the alveoli. The posteriormost foramen, medial to the fifth alveolus is much larger than the others, with a diameter equivalent to that of the second maxillary alveolus (approximately 8.5 mm). The suborbital fenestrae are elongate openings that are acuminate anteriorly and broadly rounded posteriorly. The anterior apices of the fenestrae extend anteriorly to the level of the seventh maxillary alveoli.

Dorsally the premaxilla and maxillae are separated by the nasal pair. While the midline suture between the nasals is tight and difficult to trace, the lateral sutures separating the nasal pair from the lacrimals, maxillae and premaxillae remain broadly open and obvious. The anterior tip of the nasals form a short spine that projects into the narial opening from its posterior margin and apparently prevented the premaxillae from making midline contact.

The lacrimal is a large plate of somewhat irregular shape. It bears on its dorsal surface a thick, rounded preorbital ridge that is accentuated by a broad fossa located lateroventrally to the preorbital ridge. Posteriorly the preorbital ridge becomes lower and flares transversely to encompass the anterior orbital margin. The anterior orbital margin is countersunk relative to the surface of the skull roof and as a consequence is surrounded by a broad, smooth declivous surface which extends from the prefrontal onto the lacrimal. Most of the surface of the declivity is smooth, but the lacrimal portion of it bears some irregular depressions immediately anterior to the anterior corner of the orbit. Unlike other mekosuchines, including the holotype of B. wickeni, the anterior extremity of the lacrimal is not drawn-out into a narrow point that inserts between the nasal and the lacrimal. Instead, the anterior termination is truncated, forming an irregular transverse suture with the maxilla that crosses the rounded preorbital ridge and meets the lateral margin of the nasal at a right angle.

The jugal has a long contact with the lacrimal, thus broadly excludes the maxilla from contributing to the orbital margin. It is broadly flared in lateral view at its anterior end. It constricts at the level of the postorbital process to form a compressed bar that extends under the infratemporal fenestra to contact the quadratojugal. As preserved on the left side (the right jugal bar is badly broken and incomplete), the jugal bar is dorsoventrally compressed with the ornamented external surface facing dorsally rather than laterally or dorsolaterally. Although there is some crushing in the area (shown by the near closure of the otic recess and the offset of the two ends of the postorbital bar), the orientation of the jugal bar is probably as it was in life because the same orientation is present in the well-preserved temporal fragment of NTM P8738-1 (Willis, 1997b; Fig. 15). Laterally, the anterior end of the jugal bears a tall, thick ridge that is angled ventrolaterally. The ventral margin of this ridge is undercut with deep, coalesced pits that give the ridge the form of a pendant flange as in the holotype of B. wickeni (QM F16822), the jugal fragment from Pwerte Marnte Marnte (NTM P2914-15) and a cranial fragment from D Site, Riversleigh (NTM P8738-1). The ridge abruptly diminishes in height on the maxilla and only extends for the distance of about 1 cm before disappearing entirely. A broad shallow depression is developed on the lateral surface of the jugal between the pendant ridge and the ventral margin of the orbit.

The ventral part of the postorbital bar arises from the medial surface of the jugal and is fully inset medially from its ornamented lateral surface. A gutter-like sulcus separates the postorbital process from the dorsal margin of the ornamented lateral surface. Posterior to this the jugal bar extends to the quadratojugal and forms an oblique suture with it that slopes posteroventrally. The jugal–quadrotojugal suture meets the margin of the infratemporal fenestra anterior to the posterior corner of the fenestra so that the margin of the corner is formed entirely by the quadratojugal. Unlike some mekosuchines (e.g. Trilophosuchus rackhami, QM F16856 and Mekosuchus sanderi, QM F31166) the posterior tip of the jugal lies far anterior of the quadrate condyle and the posteroventral corner of the skull.

The quadratojugal is an elongate rectangular bone present between the jugal and the quadrate. Its anterior end is damaged on both sides of NTM P91171-1, especially the parts that border the infratemporal fenestra, so no details relating to its extent or the presence of a quadratojugal spine can be discerned. There is an opening of a neurovascular foramen in the centre of the ventral surface as is typical for crocodylians and an elongate anterior process extending along the medioventral surface of the lower temporal bar.

The prefrontal is an elongate spindle-shaped bone. The surface ornament along the contact with the lacrimal is marked by a series of unusually deep pits that coalesce in places to form a broken, irregular sulcus. The orbital margin of the prefrontal is marked by a smooth declivous surface that is continuous with that of the lacrimal. Posteriorly, the prefrontal–frontal suture curves gradually laterally to meet the orbital margin, resulting in an acuminate posterior end of the prefrontal.

The frontal sends a narrow anterior process with a pointed tip between the prefrontals to meet the posterior end of the nasals. Behind the prefrontals, the frontal flares laterally and forms the posterodorsal margins of the orbits. The orbital margins are simple rounded edges that lack the smooth declivity seen on the prefrontal and lacrimal margins. Posteriorly, the frontal has an irregularly linear contact with the parietal. The frontal is excluded from contributing to the fossa that surrounds the supratemporal fenestra, although it closely approaches it with the frontal–parietal–postorbital triple junction lying on, or immediately adjacent to the outer rim of the fossa.

The crescentic postorbitals occupy the anterolateral corners of the posterior skull. The medial margin of the postorbital forms a linear margin to the supratemporal fenestra, contributing to the subtriangular shape of the fenestra. The dorsal end of the postorbital bar is formed by a descending process of the postorbital. This process is subtriangular in cross-section and is inset from the anterolateral corner of the dorsal surface both laterally and anteriorly.

In dorsal view, the lateral margins of the squamosals are angled posterolaterally relative to the sagittal line, imparting a strongly trapezoidal shape to the posterior skull table (Fig. 9). The posterolateral corner is drawn-out into acute corner with a rounded apex. The lateral surfaces of the squamosals are poorly preserved, obscuring details of the lateral sulcus for the ear flap musculature. Nonetheless, it is clear that a laterally projecting, rugose boss was not present at the posterior end of the lateral margin as it is in B. darrowi (NTM P2786-7; Figs. 15A and 15B). The squamosal margin of the supratemporal fenestra takes the form of a notch that forms the posterolateral corner of the fenestra. Few other details of the squamosals can be discerned in this specimen due to damage and poor preservation.

The parietal forms the central region of the posterior skull table. The anterolateral extremities of the parietal make contact with the postorbitals inside the supratemporal fossa. At its narrowest level the interfenestral bridge is broader between the supratemporal fenestrae than the fenestrae are long. The parietal margin of the supratemporal fenestra is concave, marking the medial corner of the subtriangular supratemporal fenestrae. Anteriorly the parietal wall surrounding the supratemporal fenestra slopes down from the dorsal rim, creating a dorsally exposed fossa surrounding the supratemporal fenestra. This fossa continues onto the postorbital. In contrast, the posterior end of the parietal wall surrounding the supratemporal fenestra is vertical and there is no dorsally exposed fossa. The cranioquadrate opening on the posterior wall of the supratemporal fenestra can be seen on the right side. It shows the anterior end of the quadrate is broadly exposed on the floor of the canal, widely separating the parietal from the squamosal under the canal as in all non-alligatoroid crocodylians (Brochu, 1997, 1999).

Posteriorly, the parietal is deeply incised by a dorsally exposed supraoccipital. Sutures are difficult to trace in this region due to damage, multiple cracks and a surface covering of thick glue, nonetheless the presence of a dorsal exposure of the supraoccipital deeply inserting into the parietal is clearly demonstrated by an isolated skull roof fragment (NTM P902-4) of B. wickeni from D Site, Riversleigh, where the bone is very well-preserved and the sutures are clear (Fig. 14). In this specimen, the parietal still contributes to the posterior margin of the skull table, although this contribution is limited to just 13% of the distance between the sagittal line and the posterolateral corner of the squamosal.

The posterior margin of the skull table of both NTM P902-4 and NTM P91171-1 bears a rounded posterior projection in dorsal view (Figs. 9 and 14), although in NTM P91171-1 it is dorsoventrally deeper and forms a knob-like tuber.

The palatines are poorly preserved, although it is clear that they widened significantly anteriorly, so that the anterior width of the pair is approximately 1.9 times their width at the palatine–pterygoid suture. The left palatine–pterygoid suture is visible and lies close to the level of the posterior margin of the suborbital fenestrae (a characteristic also present in unfigured fragments belonging to the holotype; A. Yates, 2016, personal observation), preventing the pterygoids from forming a basal section of the interfenestral strut as they do in B. darrowi (Willis, Murray & Megirian, 1990; Fig. 3C). The anterior ends of the palatines terminate bluntly posterior to the level of the anterior margins of the suborbital fenestrae.

As in all crocodylians, the ectopterygoid bears two elongated sutural contacts on each side, for articulation with the maxilla and pterygoid respectively, that are separated by a narrower waisted region. Although poorly preserved, it is apparent that the maxillary ramus of the ectopterygoid contributed to the medial margin of at least the last alveolus (more clearly preserved in NTM P8738-1; Fig. 15). The maxillary ramus continues anteriorly to the level of the ninth maxillary alveolus. A remarkable feature is that the anterior tip of the ectopterygoid is deeply inserted into the maxilla for the length of approximately two and a half alveoli (also more clearly preserved in NTM P8738-1; Fig. 15), so that a thin posterior lamina of the maxilla separates the anterior tip of the ectopterygoid from the margin of the suborbital fenestra. The ectopterygoid component of the posterolateral margin of the suborbital fenestra is evenly concave in ventral view, thus lacking the bulge seen in other crocodylians (e.g. Alligator sinensis; Brochu, 1999; Fig. 4D).

The ectopterygoid–pterygoid suture is a weakly sinuous line that extends posterolaterally from the posterolateral margin of the suborbital fenestra. It lacks the strong flexure seen in that persists through to maturity in crown-group caimans (Brochu, 1999) and is located far from the pterygoid–palatine suture. As a result the bulk of the posterior margin of the suborbital fenestra is formed by the pterygoid, an unusual condition amongst crocodylians but one shared with B. darrowi (Willis, Murray & Megirian, 1990).

The posterior median region of the pterygoid plate is depressed which tilts the opening of the choana to face anteroventrally as in globidontan alligatoroids (Brochu, 1999). The opening of the choana is shaped like the longitudinal section of an apple. As in other mekosuchines (e.g. Kambara taraina; Buchanan, 2009; Fig. 6) a low ridge extends anteriorly from each lateral margin of the choana, across the depressed area of the pterygoid. The posterior margin of the pterygoid bears a pair of well-developed posterior pterygoid processes placed on either side of the choana. In ventral view, these processes are elongate and finger-like and differ markedly from the low, blunt tubercles present in B. darrowi (A. Yates, 2016, personal observation of unfigured pterygoid fragment of the holotype, NTM P8695-8). In posterior view, the ventral peaks of the pterygoid processes are connected to the lateral margins of the occiput on either side of the median eustachian foramen by triangular webs of bone (Fig. 10). The posterior exposure of the pterygoids is dorsoventrally deeper than in most crocodylids with height that is 73% of the distance between the tips of the posterior processes (vs. 20–36% in Crocodylus porosus, Crocodylus johnstoni; A. Yates, 2016, personal observation of similarly mature specimens). Laterally, the pterygoid covers the sidewall of the braincase, dorsally to at least a point level with the lateral carotid foramen, beyond which details are not visible due to loss of most of the anterior parts of the braincase.

Neither quadrate is well-preserved or especially complete. The quadrate condyle is missing from both sides and the opening of the foramen aereum is not visible. The most remarkable aspect of the quadrate can be seen when the skull is viewed occipitally (Fig. 10). In other crocodylians, the ventromedial margin of the quadrate follows the ventral margin of the exoccipital closely so that the two bones, compressed together form an arcuate keel that extends from above the basal tuber to the opening of the foramen aereum on the main body of the quadrate near the quadrate condyle, with the suture between the two bones running along, or very close to the peak of this keel. As a consequence, the medial wing of the quadrate, is largely hidden in occipital view. In contrast, the exoccipital–quadrate suture is situated dorsal to the peak of the keel on the face of the occiput. The keel is situated entirely on a ventromedial ridge of the quadrate and as a result there is a broad exposure of the quadrate in occipital view, ventrolateral to the exoccipital.

Ventrally, a narrow rugose scar arches from the raised ventromedial ridge of the quadrate to a point close to the centre of the ventral surface. This would have allowed for the attachment of a tendon, or aponeurosis of part of the adductor musculature and is homologous with Iordansky’s crest ‘B’ (Iordansky, 1973; Fig. 10) but takes the form of a flattened to depressed scar rather than a raised crest. Poor preservation makes it impossible to see if there were further, more distal scars or crests.

A small anterolateral fragment of the right quadrate lies under the anterodorsal region of the skull table. A slender medially curving process arises from this fragment to meet the lateral process of the laterosphenoid (which is not preserved). The anterolateral surface of this process contacts the medial surface of the postorbital bar without any intervening contribution from the quadratojugal. This unusual configuration is also present in other mekosuchines (e.g. Baru sp. nov. Alcoota: NTM P5335; Quinkana timara: NMV P179632; T. rackhami: QM F16856; Kambara implexidens: QM F29662).

The exoccipitals form the dorsal and lateral margins of the foramen magnum and each extends laterally as the paroccipital process to contact the quadrate and the ventrolateral process of the squamosal. Although the midline contact of the exoccipitals is not preserved the right element reaches the level of the midline indicating that a midline contact dorsal to the foramen magnum was almost certainly present. The left side indicates that the complete foramen magnum would have taken the shape of a transversely broad ovoid. The exoccipitals project posteriorly on each side of the foramen magnum, indicating the base of a pedicel that supported an exoccipital contribution to the occipital condyle on each side. However the tips of these pedicels and the exoccipital components of the occipital condyle have broken away.

A thick crest extends from the occipital face of the exoccipital, lateral to the foramen magnum onto the paroccipital process. It partially overhangs a ventrally facing, trough-like sulcus that culminates laterally in the opening of the cranioquadrate canal. The opening of the cranioquadrate canal faces more strongly ventrally than in other crocodylians and is clearly visible as an oval foramen in ventral view (Fig. 11). The surface ventrolateral to the foramen magnum slopes gently anterolaterally and is pierced by a cluster of three foramina. The dorsomedial foramen in this group of three is the hypoglossal foramen. It is a small elongate oval opening with a maximum diameter of 3.5 mm. Ventral and slightly lateral to the hypoglossal foramen is a tiny foramen with a diameter of about 1 mm, while immediately lateral to that is a large circular foramen with a diameter of 5.5 mm. This pair of foramina can only be a divided vagus foramen. In most crocodylians, the smaller medial foramen (which conducts the glossopharyngeal and vagus nerves) opens inside the canal of the larger foramen, which conducts the communicating ramus that joins the facial and glossopharyngeal nerves (Iordansky, 1973).

Ventral to this cluster of three foramina, the exoccipital sends a tapering, acutely pointed descending process that contacts the lateral margin of the basioccipital. The tip of this process appears to reach the dorsal end of the basal tuber without contributing to the tuber itself (as in caimanines, Brochu, 1999) but damage on the left side and dislocation on the right prevent confirmation of this. Approximately halfway down the descending process, level with the ventral margin of the occipital condyle, the exoccipital is pierced by the lateral carotid foramen. This foramen is similar in size to the lateral vagus foramen and has a ventrally directed opening.

As in most other crocodylians the basisphenoid is strongly anteroposteriorly compressed so that it is visibly expressed as a thin lamina sandwiched between the pterygoids and the basioccipital. There is a dorsoventrally deep exposure of the basisphenoid sheet ventral to the median eustachian foramen. The lateral eustachian foramen is very small (less than a millimetre wide) and occurs on the basioccipital–basisphenoid suture dorsolateral to the median eustachian foramen.

The badly damaged occipital condyle was probably hemispherical when it was complete. Ventral toit, the basioccipital is deep and parallel-sided. The lateral margins are not wider than the lateral extent of the foramen magnum. Ventrally there is a median rise in the basioccipital above the median eustachian foramen, however, the peak of this rise has been broken away.

Remarks: The Pwerte Marnte Marnte site consists of densely packed, jumbled and broken vertebrate remains deposited in coarse fluviatile sediment. There is no reason to associate any of the isolated crocodylian specimens with a single individual, indeed the small size of the squamosal proves that more than one individual is present. The only likely exceptions are the angular fragments (NTM P2914-16 and 17) which were found adjacent to the larger mandibular fragment (NTM P2914-14). Given the scarcity of crocodylian remains at the site and the fact that the angular fragments are from the correct side and are of the correct size to belong to NTM P2914-14 it seems probable that they do belong even if subsequent damage prevents the fragments from fitting together. Nevertheless whether or not they do fit is a moot point because they add nothing to our understanding of the anatomy of NTM P2914-14.

Given that multiple individuals are preserved, the possibility exists that multiple crocodylian taxa are present in the Pwerte Marnte Marnte assemblage. Nevertheless, the whole sample is tentatively regarded as belonging to a single taxon. Both of the larger, more informative pieces (the jugal fragment NTM P2914-15 and the mandibular fragment NTM P2914-14) can both be referred to Baru on the basis of derived characteristics or a unique combination of characters. In the case of the jugal, the identification can be taken further for it bears a flange-like pendent jugal–maxilla ridge (Fig. 4) which is an autapomorphic character of B. wickeni (Fig. 3). The flange is present on all known B. wickeni crania from Riversleigh that preserve a jugal (NTM P8738-1, P8778-4; QM F16822) and is not present in other Baru, or indeed other mekosuchines.

The lower jaw from Pwerte Marnte Marnte (Figs. 6 and 7) can be distinguished from all mekosuchine genera except Baru by the combination of its large size, lateral ridge along the dorsal margin of the ornamented area of the surangular, lack of a longitudinal sulcus on the articular adjacent to the surangular suture and location of the medial foramen for the articular artery and alveolar nerve on the surangular–articular suture. However, there are no definite characteristics from this region of the jaw that would allow species identification.

The remaining crocodylian material from Pwerte Marnte Marnte is less diagnostic and cannot be conclusively referred to Baru. However, they are entirely consistent with Baru and do not display any character states that indicate a different taxon. Importantly, they also display plesiomorphic character states that within Baru are only known in B. wickeni. These include the squamosal fragment which lacks a laterally directed swelling on the posterolateral margin of the squamosal that is a derived characteristic that is seen in B. darrowi and an unnamed Baru species from Alcoota but not B. wickeni (Fig. 7). Similarly, the known crocodylian teeth from the Pwerte Marnte Marnte Local Fauna have smooth carinae, a plesiomorphic characteristic of B. wickeni.

Overall, all of the known crocodylian remains from the Pwerte Marnte Local Fauna can be referred to Baru, or at least show anatomical characters consistent with it. Furthermore, all of the known elements are consistent with B. wickeni and one fragment, the jugal, can be positively referred to this species on the basis of the autapomorphic jugal ridge. For these reasons, the entire crocodylian sample from Pwerte Marnte Marnte is tentatively referred to B. wickeni.

NTM P902-4 from D Site, Riversleigh is referred to B. wickeni because of its large size (it is estimated to have come from a skull about 360 mm in length) which exceeds all other crocodylians known from faunal zone A. Secondly, the median posterior semilunate process on the posterior margin of the skull table resembles that of NTM P91171-1 and is interpreted here as a likely diagnostic character of B. wickeni. Lastly, large cranial fragments of B. wickeni dominate the crocodylian assemblage from D Site and it is possible that NTM P902-4 belongs to one of the other known specimens from the site that are of similar size (e.g. NTM P8738-1).

NTM P91171-1 from 300BR, Riversleigh can be referred to B. wickeni on the basis of the autapomorphic pendant jugal–maxilla flange and the presence of the following character states that distinguish B. wickeni from B. darrowi and the unnamed Baru from Alcoota: the nasals extend to the margin of the external naris; there is a strongly developed preorbital ridge on the lacrimal that is flanked by a lateral depressions; there is no posterolateral squamosal boss; there are wide gaps between the sixth, seventh and eighth maxillary teeth (admittedly not as wide as in other B. wickeni specimens but still wider than in the similarly mature holotype specimen of B. darrowi); the palatine–maxilla suture is gently rounded; palatine–pterygoid suture level with posterior margin of suborbital fenestra; the posterior pterygoid processes are elongate, finger-like projections in ventral view.

NTM P91171-1 is one of the most complete Baru skulls known and is certainly the most complete known for B. wickeni. It fills in many details of anatomy that were previously unknown and allows for a revision of the original diagnosis for the species (Willis, 1997b). One of the main distinguishing features stressed in the original diagnosis and description of the species was its apparent narrower snout with a pointed rostral tip. However, the snout proportions of NTM P91171-1 are almost the same as those of the holotype of B. darrowi (NTM P8695-8), which is from a similarly mature individual. This suggests that the narrower snout of the holotype of B. wickeni might be a feature relating to its smaller size and presumably younger ontogenetic stage. Similarly, the premaxillary tip of NTM P91171-1 is as bluntly rounded as the holotype of B. darrowi or mature specimens of the undescribed Baru from Alcoota (e.g. NTM P6319) suggesting that the slightly more pointed premaxillary tip of the holotype (QM F16822) is another characteristic of its presumably young age. A third character used to establish B. wickeni as distinct from other species was the strong constriction of the anterior nasals between the premaxilla. This is visibly reflected in the marked angulation of the lateral margin of the nasals at the maxilla–premaxilla–nasal triple junction. These angulations are only mildly developed in NTM P91171-1 and do not differ appreciably from the anterior nasals of QM F30319 (referred to B. darrowi) indicating that the more strongly angled nasals of QM F16822 are a feature of individual variation and not a diagnostic characteristic of the species. Thus, it can be seen that the new specimen of B. wickeni casts doubt on the original characteristics used to establish B. wickeni as a species distinct from B. darrowi. Nevertheless there are numerous other characteristics, given in the diagnosis above, that do appear to differ markedly between the species and allow B. wickeni to continue to be recognised as a valid taxon.

Baru darrowi Willis, Murray & Megirian, 1990

(Figs. 1A, 3A, 16A, 16B, 17–18)

Figure 17 Baru darrowi Willis, Murray & Megirian, 1990, QM F30319, snout fragment of a juvenile including left premaxilla and nasal from Ringtail Site, Riversleigh World Heritage Area.

(A) Dorsal view. (B) Ventral view. (C) Lateral view. (D) Medial view. 1–4, premaxillary alveoli; en, external naris; dps, dorsal premaxillary symphyseal surface; if, incisive foramen; ma, articular surface for attachment of the maxilla; n, nasal; na, articular surface for attachment of the nasal; nc, narial canal; pdp, posterior dorsal process of the premaxilla; pps, palatal premaxillary symphyseal surface; rn, reception notch for fourth dentary tooth; rp, reception pit for first dentary tooth. Scale bar = 20 mm.

Figure 18 Baru darrowi Willis, Murray & Megirian, 1990, QM F31185, left maxilla of a juvenile from Ringtail Site, Riversleigh World Heritage Area.

(A) Lateral view. (B) Medial view. (C) Dorsal view. (D) Ventral view. appa, articular surface for the attachment of the anterior process of the palatine; eca, articular surface for the attachment of the ectopterygoid; fan, foramen for n. alveolaris dorsalis caudalis; ja, articular surface for the attachment of the jugal; la, articular surface for the attachment of the lacrimal; na, articular surface for the attachment of the nasal; nc, narial canal; pma, articular surface for the attachment of the premaxilla; rp, reception pit for dentary teeth; sym, symphyseal surface. Scale bar = 20 mm.

Revised diagnosis: Distinguished from B. wickeni by: median sutural contact of premaxillae posterior to external naris; weakly developed preorbital ridge on lacrimal; low ridge extending from anterior end of jugal onto posterior end of maxilla; squamosal with laterally projecting, posterolateral boss; sixth, seventh and eighth maxillary teeth separated by gaps less than length of preceding alveolus in adults; acutely triangular anterior process of palatines; palatine–pterygoid suture anterior of posterior margin of suborbital fenestra; anterior tip of ectoptergoid inserted into maxilla for short distance (less than length of tenth maxillary alveolus);posterior pterygoid processes are low, blunt tubercles in ventral view; dorsal posterior lobe of dentary symphyseal surface overhangs ventral lobe; splenial closely approaching, or contacting symphyseal surface; larger teeth with minutely crenulated carinae.

Type Locality: Blast Site, Bullock Creek, Northern Territory. Camfield Beds, Bullock Creek Local Fauna, middle Miocene.

New material: QM F30319, a left premaxilla with an articulated fragment of the left nasal (Fig. 17). QM F31013, a fragment of left maxilla. QM F31185, a complete left maxilla (Fig. 18).

Locality and stratigraphic age of the new material: Ringtail Site, Riversleigh WHA, Queensland. Unnamed calcareous pool deposit, Ringtail Local Fauna, Riversleigh faunal zone C, radiometrically dated to 13.56 ± 0.67 ma, middle Miocene (Woodhead et al., 2016).

Description of the Ringtail Site specimens: The premaxilla (Fig. 17) is complete, save for the anteriormost surface and the second premaxillary tooth. Its absolute ontogenetic age cannot be determined but it is presumed to be a young juvenile because it is less than 40% of the linear dimensions of the mature holotype individual. The clearly open sutures and the separation of the nasal symphysis support this interpretation. Despite the missing surface it is apparent that, like other Baru, the anterior surface was both deep and almost vertical. The external naris faces dorsally and when complete would have been almost exactly as wide as it was long (determined by mirror imaging the complete left side of the opening) in the shape of a rounded trapezoid. The premaxillae completely surround the external naris and there is a short medial sutural surface behind the naris where the opposite premaxilla would have contacted its partner. The premaxillary pair sends a short anterior median spine that protrudes into the opening of the external naris for a distance of 3.5 mm. On the ventral surface of this spine there is a sutural contact for the anterior tip of the nasal (which is missing due to breakage). This contact indicates that the nasals also reached the naris, albeit ventral to the superficial premaxillary cover and invisible in dorsal view. Ventrally the premaxilla bears four alveoli. There is a broad gap between the first and second alveolus indicating that, as in other crocodylians with four premaxillary teeth (such as in many mature individuals of Crocodylus porosus, e.g. NTM R12638; Brown et al., 2015), it is the primitive second tooth that is missing (Brochu, 1999). It is interesting to note that the loss of this tooth would have had to have occurred early in ontogeny if the specimen is a young juvenile as argued above. The horizontal premaxillary plate curves ventrally adjacent to the alveoli to form a lingual alveolar wall. The wall bulges lingually around the alveoli while narrow, lingually facing depressions occur between alveoli two and three as well as three and four. A circular reception pit for the first dentary tooth occupies the space on the premaxillary palate between the first premaxillary alveolus anteriorly, the send premaxillary tooth laterally and the incisive foramen laterally. The floor of the reception pit is complete and separates it from the narial cavity, unlike the old adult holotype of B. darrowi where the pit has merged with the incisive foramen. Whether this difference is due to post-mortem damage to the holotype or absorbtion of bone during its life is not known. An irregularly spaced line of neurovascular foramina opens along the base of the alveolar wall. Mirror imaging indicates that the complete incisive foramen would have been broadly lanceolate with an elongated anterior point. The margins of the premaxillary symphysis anterior to the incisive foramen are not distinct so it is hard to judge exactly how far anteriorly the foramen extended but it is clear that the anterior end closely approached the lingual margins of the first premaxillary alveoli, if not actually abutting them. It is, however, clear that the incisive foramen did not intrude between this pair of alveoli. The posterior margin of the incisive foramen is level with the third, and penultimate, premaxillary alveolus. The anterior half of a ventrolaterally facing notch for receiving the fourth dentary tooth occurs between the fourth premaxillary alveolus and the maxillary suture. This notch is bounded dorsolaterally and ventromedially by thin, low and sharp ridges. The palatal premaxilla–maxilla suture is oriented medially in a roughly linear transverse line.

The maxilla (QM F31185; Fig. 18) has already been described by Willis (2001), so only a few salient points will be mentioned here. The number of maxillary alveoli cannot be observed directly because the region between the anterior and posterior alveolar convexities is crushed and the alveoli obscured. However, the pattern of alveoli in this region is stable within Baru and the medial reception pits, which align with the gaps between alveoli, can also be observed. Thus, it is clear that the crushed region is obscuring maxillary alveoli six and seven, while the first clear alveolus posterior to the crushed region is the eighth maxillary alveolus. As in other Baru (e.g. NTM P91171-1, P8695-8, P8738-1; Fig. 15), the eighth alveolus remains close to, but separated from the closely spaced, enlarged alveoli of the posterior convexity (alveoli 9–11). Counting from alveolus eight, it is clear that QM F31185 had 14 alveoli, which is one more than the holotype of B. darrowi (NTM P8695-8). While the modal number of maxillary teeth in modern crocodylians is apparently constant within species and does not vary ontogenetically, deviations of one, or occasionally two, alveoli from the mode do occur in 20–30% of individuals within a species (Brown et al., 2015) and are not indicative of a taxonomic difference. The lateral wall of the narial canal (visible when the maxilla is viewed medially) is smooth and lacks any recesses. Also visible in medial view is the foramen for the posterior branch of the dorsal alveolar nerve. The opening occurs on the medial side of the dorsal lamina, above the alveoli and level with the anterior end of the ectopterygoid articulation. Unlike most other crocodylians, but as in other Baru (e.g. NTM P912; Fig. 2A), the opening of this foramen faces dorsally and lies close to the dorsal edge of the maxilla. The sutural contact with the palatine is well preserved. It indicates that, like the holotype of B. darrowi (NTM P8695-8), the palatine–maxilla suture formed a broad chevron with the pointed apex directed anteriorly. The tip of the anterior process extends for a short distance anterior to the anterior margin of the suborbital fenestra and reaches approximately level with the seventh alveolus. Posteriorly the articular scar for the attachment of the ectopterygoid is well preserved. It indicates that the anterior tip of the ectopterygoid inserted into a notch in the maxilla and was separated from the lateral margin of the suborbital fenestra by a short medial lamina of the maxilla. This is a derived condition seen in a number of mekosuchines including B. wickeni (NTM P8738-1; Fig. 15), ‘Baru’ huberi (QM F31063), Mekosuchus sanderi (QM F3118), ‘Pallimnarchus’ gracilis (QM F1752) and Kambara implexidens (QM F29662).

Remarks: The deep, near vertical anterior profile of the premaxilla and the presence of just four premaxillary teeth at an early ontogenetic stage are derived characters that allow QM F30319 to be referred to Baru. The median premaxillary contact behind the external naris is an autapomorphic character that allows referral to B. darrowi.

Four other specimens from Ringtail Site were referred to Baru sp. by Willis (2001). Of these, two of the maxillae (QM F31013, F31185) can be referred to Baru on the basis of the combination of the following characters: rounded alveoli, well-developed maxillary alveolar convexities, anterior tip of the ectopterygoid inserting into the maxilla and separated from the lateral margin of the suborbital fenestra; medial foramen for the posterior branch of the maxillary nerve opens dorsally near the dorsal edge of the maxilla (Fig. 9B). The latter character has not yet been described but is an apparent synapomorphy of Baru that is present in the holotype of B. darrowi (NTM P8695-8) and an undescribed species from Alcoota (NTM P912; Fig. 2A) but not in other crocodylians including Kambara implexidens (QM F29662), Quinkana meboldi (QM F31056), Mekosuchus kalpokasi (Mead et al., 2002; Fig. 3B), Crocodylus novaeguineae (QM J5332) and Alligator mississippiensis (NTM R36716) (Figs. 2C and 2D). Note that the foramen being referred to here is not the main foramen for the dorsal alveolar nerve and associated vessels which enters the maxilla dorsal to the apex of the suborbital fenestra and immediately lateral to the aperture for the caviconchal recess. The foramen in question allows passage of a branch of the maxillary nerve (cranial nerve V2) that splits from the dorsal alveolar nerve before the latter enters the maxilla near the apex of the suborbital fenestra. This posterior branch of the maxillary nerve can be clearly seen in George & Holliday (2013; Fig. 3B) and is also illustrated in Witmer (1995; Fig. 14). Few workers have paid attention to this foramen nevertheless it is a constant feature and I have observed it in every crocodylian. I have examined where the internal surface of the maxilla can be clearly seen. Iordansky (1973; Fig. 14E) figured the foramen and labelled it as a pneumatic foramen. However the canal joins the alveolar canal for the dorsal alveolar nerve and does not communicate with the antorbital sinus (A. Yates, 2016, personal observation of unregistered Crocodylus johnstoni in NTM collections).

The maxillae can be specifically referred to B. darrowi for several reasons. Firstly, they co-occur with QM F30319 (itself referrable to B. darrowi) and share very similar reciprocal sutural surfaces indicating that they are likely from the same taxon. Secondly, the notch for reception of the anterior tip of the ectopterygoid is shallow, unlike the deep notch and associated medial lamina of B. wickeni (Fig. 15) but much the same as in maxillae of B. darrowi from the type locality (e.g. NTM P87103-12). Thirdly, and most importantly, the Ringtail Site maxillae share a short, pointed triangular anterior termination of the palatine pair that that is level with the seventh maxillary tooth with the holotype of B. darrowi (NTM P8695-8). This triangular anterior termination differs markedly from the bluntly rounded and posteriorly placed palatine termination of B. wickeni (Fig. 11) and the undescribed Baru species from Alcoota (e.g. NTM P5335) and may well be another autapomorphy of B. darrowi. The processes of the Ringtail maxillae are not preserved themselves but their shape and position can be clearly deduced from the articular surfaces on the maxillae. A comparison of QM F31185 with the original illustrations of the holotype of B. darrowi (Willis, Murray & Megirian, 1990; Fig. 1C) would seem to indicate that the palatines of the latter were less anteriorly produced, only reaching the level of anterior margin of the eighth maxillary alveolus and not protruding beyond the anterior ends of the suborbital fenestrae. However, the anterior ends of the palatines of NTM P8695-8 are damaged with a broad crack extending roughly parallel to the palatine–maxilla suture. The original figure fails to capture this damage and the palatine–maxilla suture is drawn following the posterior margin of the crack (Fig. 19). Close examination of the specimen indicates that the sutural surface is actually represented by the anterior margin of the cracked zone and that the anterior end of the palatine pair drew level with the seventh maxillary tooth and protruded anterior to the suborbital fenestrae as in QM F31185 (Fig. 19).

Figure 19 Baru darrowi Willis, Murray & Megirian, 1990, NTM P8695-8, anterior palatal region of holotype skull in ventral view showing differing interpretations of the maxilla-palatine suture.

(A) photograph. (B) photograph overlain with sutural interpretations. Black lines indicate uncontroversial sutures and structures, dark green line represents the maxilla-palatine suture as interpreted in this paper (note that the anterior palatine process extends anterior to the anterior margin of the suborbital fenestra), blue line represents the path of the maxilla-palatine suture as illustrated in Willis, Murray & Megirian, 1990 (Fig. 1), red line indicates the approximate path of the maxilla-palatine suture in QM F31185. Abbreviations: mx, maxilla; pal, palatine; sof suborbital fenestra. Numerals indicate the maxillary alveolus number. Scale bar = 50 mm.

The dentary (QM F31004; Fig. 20) that Willis (2001) referred to Baru presents a problem. Like B. darrowi it does possess an overhanging posterior dorsal lobe of the dentary symphyseal surface but unlike that species the anterior tip of the splenial is widely separated from the symphysis. The specimen also differs from all other known specimens of Baru in being much smaller, and in having a highly dorsoventrally compressed symphysial platform. Perhaps all of these discordant features are the result of an extremely early ontogenetic stage, though the anterior position of the splenial has not been documented to be an ontogenetically variable character in crocodylians. An alternative explanation is that the dentary actually belongs to T. rackhami, a very small mekosuchine that is also known from Ringtail Site (Willis, 1993). Willis (2001) suggested that yet another jaw form from Ringtail Site, the ‘robust dentary form’ as he called it, could possibly represent the lower of T. rackhami. New, more complete, specimens of this taxon are currently being studied by the author and they are neither Trilophosuchus nor Baru. Unlike the ‘robust dentary form’, QM F31004 is the right size to fit T. rackhami. Furthermore the dorsoventrally compressed symphyseal platform matches the compressed platforms seen in the related dwarfed mekosuchine genus, Mekosuchus. Only further discoveries, either of T. rackhami with lower jaws or more complete, highly juvenile Baru can solve this question.

Figure 20 Mekosuchinae gen. et. sp. indet., QM F31004, right dentary from Ringtail Site, Riversleigh World Heritage Area.

(A) Lateral view. (B) Medial view. (C) Dorsal view. (D) Ventral view. atsa, anterior tip of the articular surface for the splenial; emf, notch forming the anterior margin of the external mandibular fenestra; mc, Meckelian canal; saa, articular surface for the surangular; sym, symphyseal surface of the dentary. Scale bar = 20 mm.

Discussion

B. darrowi has been recorded from Riversleigh WHA before (Willis, Murray & Megirian, 1990), but these specimens, which hail from faunal zone A, are now referred to B. wickeni (Willis, 1997b). QM F30319 and F31185 from Ringtail Site indicate that B. darrowi was present in faunal zone C of the Riversleigh WHA. Fortunately, Ringtail Site is one of the Riversleigh sites for which a radiometric date could be obtained (Woodhead et al., 2016). This date of 13.56 ± 0.67 ma, places the site approximately on the boundary of the Langhian and Serravallian stages in the middle Miocene (Woodhead et al., 2016). Bullock Creek, the type locality for B. darrowi, shares a number of mammal species with faunal zone C, particularly the younger interval zones within zone C. Among the shared species are Mutpuracinus archibaldi Murray & Megirian, 2000, Wakaleo vanderleuri Clemens & Plane, 1974 and Neohelos stirtoni Murray et al., 2000. These marsupials are only known from younger faunal zone C deposits of the C2 and C3 intervals (Archer et al., 2006; Arena et al., 2015). Although the mammalian fauna of Ringtail Site lacks critical species that would allow it to be placed securely in an interval zone within faunal zone C (Arena et al., 2015), the radiometric date obtained from this site was one of the younger ones for faunal zone C. Therefore, Ringtail Site, like the Bullock Creek LF, most likely correlates with the upper part of faunal zone C. Thus the occurrences of B. darrowi in the Northern Territory and Queensland are close to coeval. It is now apparent that B. darrowi was a widespread species in Northern Australia around 13–14 ma (middle Miocene), stretching from at least Bullock Creek in the west to Riversleigh in the east, a distance of approximately 800 km (Fig. 21).

Figure 21 Geographic and stratigraphic distribution of Baru darrowi Willis, Murray & Megirian, 1990 and B. wickeni Willis, 1997a.

Map of Australia showing Queensland (Qld) and the Northern Territory (NT) and the late Oligocene and middle Miocene sites that have produced these species. Simplified biochronological column for the Riversleigh WHA sequence of Oligo-Miocene rocks, highlighting the known occurrences of Baru. FZ, faunal zone.

The presence of B. wickeni in the Pwerte Marnte Marnte LF is the first record for the species in the Northern Territory and indeed the first record outside of the Riversleigh WHA. The location of the Pwerte Marnte Marnte LF in central Australia not only represents a westward range extension but also a significant southerly extension, suggesting that its potential range may have encompassed the entire northern half of the continent.

A significant detail of the Pwerte Marnte Marnte deposit is that it lies on the northern fringe of the area of sedimentary cover laid down in the Lake Eyre Basin (LEB) during its late Palaeogene–early Neogene depositional phase (Megirian et al., 2004; Murray & Megirian, 2006; Habeck-Fardy & Nanson, 2014; Fig. 22). Willis (1997a) noted that the well-sampled Etadunna Formation in the LEB of central South Australia contained abundant crocodylian remains attributable to the mekosuchine Australosuchus clarkae Willis & Molnar, 1991 but not a trace of any of Baru, or indeed any of the other the mekosuchines found at Riversleigh. Conversely, no trace of Australosuchus has ever been recovered from the crocodylian-rich late Oligocene deposits of Riversleigh (Willis, 1997b). Time differences cannot explain this pattern because the upper part of the Etadunna sequence (Ngapakaldi and Ngama Local Faunas) correlates with faunal zone A of Riversleigh (Archer et al., 1989; Myers & Archer, 1997; Travouillon et al., 2006; Arena et al., 2015) and yet remain rich in Australosuchus clarkae. Willis (1997b) suggested that this distributional disparity was the result of aquatic taxa such as Australosuchus and Baru being restricted to separate and isolated drainage systems. The presence of B. wickeni at the northern end of the LEB potentially falsifies this hypothesis but the tectonic evolution of the Australian continent needs to be considered first, given that it has wrought many changes to hydrological basins and their depocentres throughout the Cenozoic (Sandiford et al., 2009).

Figure 22 Geographic distribution of large mekosuchines in the late Oligocene of Australia.

Map of Australia showing the occurrences of the coeval Australosuchus clarkae Willis & Molnar, 1991 and Baru wickeni Willis, 1997b and position and approximate extent of Lake Eyre Basin (pale brown) in the Oligo-Miocene (modified from Callen et al., 1986; Murray & Megirian, 2006). The north/south division of B. wickeni and A. clarkae is marked by the 26°S line of latitude (approximately 43° S palaeolatitude in the late Oligocene).

The modern hydrological LEB is far larger than its late Paleogene–early Neogene geological basin, covering almost 15% of the continent and encompasses the present day Pwerte Marnte Marnte site. The site falls in the present day catchment of the Finke River, one of seven major catchments of the modern hydrological LEB (Habeck-Fardy & Nanson, 2014). The question is, was this the case at the time of deposition of the Pwerte Marnte Marnte LF, or did the area drain to the north or northeast, eventually connecting with the Karumba Basin where the Riversleigh WHA lies? The latter scenario seems unlikely and all palaeodrainage reconstructions for the region show the Finke and adjacent Todd Rivers systems draining to the southeast into the LEB for the duration of the Cenozoic (e.g. Edgoose & Ahmad, 2013). It is pertinent to note that there are (and were) highlands of the Arunta Region (Aileron and Warumpi Provinces) to the north and northeast (Ahmad & Scrimgeour, 2013) that were were shedding sediments to the south in the early Cenozoic, forming a piedmont of coalesced alluvial fans along its southern margin which are now present as dissected mesas (Megirian et al., 2004). Furthermore there are several small intermontane basins developed within the Aileron Province (e.g. Ti Tree Basin and the Waite Basin) which have been accumulating sediment since the Mesozoic in some cases (Edgoose & Ahmad, 2013). These indicate that the Pwerte Marnte Marnte site and Riversleigh were separated by multiple drainage basins in the mid-Cenozoic as they are now and that B. wickeni was more than capable of dispersing across these basins.

What does continue to segregate the occurrences of Australosuchus clarkae and B. wickeni is latitude, with the former not known north of 27° S and the latter not found south of 25° S. This suggests that it was palaeolatitude, and hence palaeoclimate, that separated these taxa, rather than access to drainage systems. Indeed, the most southerly record for Australosuchus clarkae, Lake Pinpa, lies at 31° S and would have had a palaeolatitude between 45° and 50° S in the Oligocene (from McGowran et al., 2004; Fig. 1). This exceeds the highest latitude obtained by a viable population of an extant crocodylian (36° N in Alligator mississippiensis, from distribution given in Neill (1971)). Furthermore, the Oligocene represents a cool period in the saw-tooth history of Australian palaeotemperatures (McGowran et al., 2004), indicating that Australosuchus clarkae may have been an unusually cold-tolerant crocodylian. This may explain the why Australosuchus clarkae is the sole crocodylian from the Etadunna Formation while adequately sampled crocodylian assemblages from the Oligo–Miocene of Northern Australia usually contain multiple species.

The age of the Pwerte Marnte Marnte LF is a matter of some uncertainty. When the local fauna was first reported, Murray & Megirian (2006) determined that it was closest to, but nonetheless predated, the basal-most local faunas of the Etadunna and Namba Formations which date to the late Oligocene (Woodburne et al., 1994). Murray & Megirian (2006) reached this conclusion based upon the apparent absence of any described species from other late Oligocene sites and the apparent plesiomorphic nature of the unnamed ilariid from the Pwerte Marnte Marnte LF. ‘Stage-of-evolution’ biochronology indicates that the age of the Pwerte Marnte Marnte LF is a greater than the oldest local fauna of the Namba Formation (Pinpa Local Fauna) which contains a more derived ilariid, Ilaria ilumidens. Murray & Megirian (2006) also noted that the fauna must postdate the basal radiation of diprotodontan marsupials given the presence of diprotodontan subclades such as Ilariidae, Wynyardiidae, Diprotodontidae, Macropodoidea, Phalangeroidea and Petauroidea. Megirian et al. (2010) upheld Murray & Megirian’s (2006) correlation when establishing a series of Australian Land Mammal Ages. The basal local faunas from the Etadunna and Namba Formations were placed in the Etadunnan Land Mammal Age while the Pwerte Marnte Marnte LF was regarded as pre-Etadunnan (Megirian et al., 2010).

In contrast, Black et al. (2012) suggested that the Pwerte Marnte Marnte LF could be correlated with Riversleigh’s faunal zone A, on the basis of the shared presence of the unusual marsupial Marada arcanum. As Riversleigh’s faunal Zone A correlates with the Ngapakaldi and Ngama Local Faunas from higher in the Etadunna sequence (based on the shared presence of the marsupials Ngapakaldia bonythoni and Kuterintja ngama) this would place the Pwerte Marnte Marnte at a later time in the Oligocene, close to the Oligo–Miocene boundary (Woodburne et al., 1994). Regardless of how the Pwerte Marnte Marnte LF is correlated there is no disagreement that it is Oligocene, and probably late Oligocene in age. Thus, as in the younger B. darrowi, B. wickeni appears to have been a geographically widespread species with known occurrences falling within a relatively narrow timespan, in this case the late Oligocene. One immediately useful aspect of this biochronological information is the placement of 300BR, a Riversleigh site that could not be placed within the Riversleigh faunal zones due to a lack of informative mammal fossils (Travouillon et al., 2006). As described above NTM P91171-1 is a skull collected by D. Megirian from 300BR that is referrable to B. wickeni on the basis of many characteristics outlined above. Based on the presence of B. wickeni, the site 300BR can now be resolved as belonging to faunal zone A.

The chronology of Australia’s Cenozoic terrestrial vertebrates has been a particularly vexing problem that is exacerbated by the prevalence of spatially and temporally isolated assemblages that present few opportunities for lithostratigraphic correlation using superposition (Megirian, 1994). Furthermore, the stable intraplate position of the continent has not been conducive to volcanic activity and as a result only a few sites can be dated absolutely with radiometric techniques (Black et al., 2012; Woodhead et al., 2016). For these reasons, palaeontologists have relied heavily, indeed almost exclusively, on biochronological methods, particularly stage-of-evolution biochronology (Stirton, Woodburne & Plane, 1967; Megirian, 1994) or seriation based on taxon presence or absence (Travouillon et al., 2006; Megirian et al., 2010), as the main means placing these assemblages in a relative timescale. To date, this work has been carried out exclusively with marsupial fossils. In particular, certain marsupial clades, e.g. zygomaturine diprotodontids, wakaleonine marsupial lions and stem bandicoots of the genus Yarala amongst others show fairly regular taxonomic turnover through the sequence of sites preserved in the Riversleigh WHA and elsewhere in Australia (Stirton, Woodburne & Plane, 1967; Megirian et al., 2010; Arena et al., 2015). The turnover from B. wickeni in the late Oligocene to B. darrowi in the middle Miocene and Baru sp. nov. in the late Miocene seems to form another such lineage that can be of use for ordering terrestrial vertebrate deposit, especially where biostratigraphically significant marsupials have not been found (such as at 300BR). Hopefully, future work will throw light on Baru from the early Miocene which currently represents a large gap in the record of the genus. Other mekosuchine genera, e.g. Quinkana and a presently undescribed genus, show similar taxonomic turnover and these will be explored in future publications.

Conclusion

The new Baru fossils from Queensland and the Northern Territory extend the range of B. darrowi from the Northern Territory into Queensland and vice versa for B. wickeni, indicating that these species had broad geographic ranges. The influx of new anatomical information, especially for B. wickeni, allows the diagnoses for both species to be refined. All diagnosable specimens of B. darrowi are restricted to middle Miocene sites and all specimens of B. wickeni from correlatable sites date to the late Oligocene. This suggests that the species of Baru were temporally restricted and underwent taxonomic turnover across their geographic range. Dating Australian terrestrial vertebrate fossil sites has proved difficult and is still largely based on certain clades of marsupials that show taxonomic turnover (described as evolutionary lineages, Arena et al. (2015)) combined with seriation analysis of presence/absence data of marsupial taxa (Travouillon et al., 2006; Megirian et al., 2010). The addition of mekosuchine clades to these data can only enhance such analyses and can be especially helpful in placing marsupial poor sites (such as 300BR in the Riversleigh WHA) into a biostratigraphic framework. Baru is not unique amongst Mekosuchinae in the respect and other long-ranging mekosuchine genera are likely to yield similar patterns of widespread species turnover on a pace similar to marsupial taxa, after taxonomic revision.

The occurrence of B. wickeni in the extreme northern margin of the geological LEB in the southern Northern Territory is significant on biogeographic grounds for it falsifies the hypothesis that the Oligo–Miocene mekosuchine faunas of Australia were endemic to major drainage basins (Willis, 1997a).

I thank Kristen Spring and Scott Hocknull of the Queensland Museum for facilitating access to the collection in their care and for organising a loan of specimens from that collection. Trevor Worthy provided useful suggestions for improvement of the preprint version which have been used here. The manuscript was greatly improved by thorough reviews from Michela Johnson, Steven Salisbury, Lucy Leahey and an anonymous reviewer.

Institutional Abbreviations

NMV Museum Victoria, Melbourne, Victoria, Australia

NTM Museum and Art Gallery of the Northern Territory, Darwin and Alice Springs, Northern Territory, Australia

QM Queensland Museum, Brisbane, Queensland, Australia.

Additional Information and Declarations

Competing Interests

Author Contributions

Data Availability

The author declares that he has no competing interests.

Adam M. Yates conceived and designed the experiments, performed the experiments, analysed the data, contributed reagents/materials/analysis tools, wrote the paper, prepared figures and/or tables, reviewed drafts of the paper.

The following information was supplied regarding data availability:

The raw data is included in the manuscript.

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
