# Peer review of "The biochronology and palaeobiogeography of Baru (Crocodylia: Mekosuchinae) based on new specimens from the Northern Territory and Queensland, Australia"

_PeerJ, doi:10.7717/peerj.3458_

## Round 0.1 · original submission · Major Revisions

Dear author,

Very sorry for the lateness of this review. The delay was due to SVP, and a third reviewer who has not submitted their review. I have decided not to wait, and return this now. I have accepted the decision of 'major revision' from both reviewers.

Please read reviewer two's comments regarding biochronology. Are crocodylians really a good biochronological marker? I think this requires some serious thought before returning the manuscript.

·

Basic reporting

The general outlook of the article is good. There are, however, some inconsistencies. Sometime specimen numbers are written out (e.g. ...an undescribed species from Alcoota (NTM P912)...), but other times they are absent. This is similar when referring to figures; sometimes a figure is referred to, but other times, when an important specimen or feature is mentioned, there will be no figure reference. The grammar (scientific wording, use of commas, hyphens, etc.) is is largely inconsistent and used somewhat poorly in the first 7 pages of the paper, then abruptly become much better (the description of QM F31185 maxillae and onward). The paper definitely needs to be looked over and fixed in terms of grammar, sentence structure, and consistency with the inclusion of specimens/specimen numbers/figures.

Experimental design

The research and data presented in the article are interesting, especially the geographic distributions in the discussion. The descriptions, while decent, could be clearer. Use '&' when referencing three authors in text, as it looks less wordy.

Validity of the findings

There is no institutional abbreviations or conclusions section. These would be beneficial to have. Especially a conclusions section, in which the author can summarize the findings, highlight their significance and why it is important, and offer future endeavors with this data. The geographic distribution findings are interesting, yet it is not emphasized why they are important.

Additional comments

It's definitely on the right track! It was quite interesting. I would fix what I've mentioned above, and see comments on the attachment.

Reviewer 2 ·

Basic reporting

n/c

Experimental design

n/c

Validity of the findings

1. I buy the referral of new material to B. darrowi convincing, but not the new B. wickeni. It simply cannot be referred to a species. I’ve now spent quite a bit of time staring at Figure 2, comparing it with the original description of the species and the notes I took of the material when I saw it several years ago, and I just don’t find anything diagnostic about it. I would recommend referring this material to Baru sp. – anything more precise is not sustainable.

2. It isn’t clear what, exactly, the authors are arguing about biochronology. Based on what I’ve observed in several sequences, croc species may not be as long-lived as was thought in the past, but they can still persist (as morphologically recognized) for well over a million years. Some may extend for as many as three million. So if the authors are claiming that the presence of B. darrowi puts a unit close to 13.5 Ma, that’s ok – just so long as they’re willing to acknowledge that the unit could be as old as 15 Ma or as young as 11 Ma. Much depends on what we know about rates of evolution for this particular group of crocodylians, and we don’t know much at all.

Additional comments

A couple of other comments:

Sentences sometimes start with a generic abbreviation (e.g. line 42, “B. huberi Willis, 1997a is a….”) Genus names should always be spelled out when they start a sentence.

Line – 265 – having seen both inexpectatus and kalpokasi, the implied synonymy is puzzling, since they’re demonstrably different.

Reviewer 3 ·

Basic reporting

See below

Experimental design

See below

Validity of the findings

See below

Additional comments

Well written, good figures, interesting findings and propositions re the taxonomy and temporal and palaeogeographic distribution of Buru material.

Things to address

1. Use crocodylian throughout rather than ‘crocodilian’. The former has a more precise phylogenetic meaning than the latter.
2. Provide an amended genus diagnosis for Baru.
3. Include complete lists of holotype, revised paratypes and referred material, highlighting new referrals (for both B. wickeni and B. darrowi). There is a lot of material referred to Baru, at least three recognised species, and a lot of the material from Riversleigh is spread between two institutions (Alice Springs and Brisbane). This makes it all very confusing. Some updated listings (and ideally a table in the text or as part of an appendix) would be highly desirable.
4. Refine the wording of various descriptive phrases in the diagnoses and osteological descriptions. Describe the state that is present, not by what is absent. Avoid ambiguous terms such as ‘widely spaced’,
4. Taxonomic referrals (BIGGEST ISSUE)

You don’t really say anything about the Pwerte Marnte Marnte fossil assemblage. How is the croc stuff from this assemblage preserved? Is it all associated? Articulated, isolated? How many individuals seem to be present? Do you have any reason to believe that there is more than one taxon represented? Is the material that is described here new or was it the material mentioned by Murray and Megerian (2006)? All this should be made clearer in the intro I think.

How is the angular NTM P2914-16, 17 assigned to B. wickeni if there are not diagnostic characteristics on this bone (this part of the mandible is not preserved on the holoytpe from White Hunter site)? You’ve left other things out (e.g., osteoderms, teeth, vertebrae, etc) because you considered them ‘undiagnostic’. How does this not apply to the angular? Similarly, with the squamosal fragment, the absence of a derived trait seen in other taxa does not qualify as a diagnostic (=apomorphic) condition. The condition seen on this squamosal is plesiomorphic for most crocodyloids.

From what I can see, the referral of the Pwerte Marnte Marnte material to Baru wickeni pretty much all hinges on the jugal fragment (NTM P2914-15; incidentally, this is the same number as the angular, are they from the same individual?) Is the jugal fragment enough to substantiate this referal? Could that flange on the jugal be a feature that is shared with B. wickeni, such that this is a different species, closer to B. wickeni than B. darrowi, but distinct in its own right? It seems just as likely given its occurrence in the NT.

Assignment of the Ringtail Baru sp. material to B. darrowi. Maxilla (QM F31185). You focus a lot on what you refer to as the foramen for the n. alveolaris dorsalis caudalis (I’m not sure why this particular term is Latinized, where others in the description are not, and why caudal is used in preference to posterior, given you have gone with anterior and posterior throughout). Based on how it is described and from what you have labeled in the figure, I don't think this is the aperture for the dorsal alveolar nerve and vessels. It doesn't look like it on your figs (but hard to tell because the maxilla is a bit squashed in the area where it should be). The aperture that you are referring to typically passes dorsolateral to the paranasal sinus cavitites (caviconchal recess) in crocodyloids and laterally in alligatoroids (see attached photos). It is never medial to the coniconchal sinus as you have figured it. It is usually lateral to it and hugging the maxillary tooeth row. What you have labelled looks more like a secondary (medial) aperture associated with the caviconchal sinus. With regard to the inferred nature of the anterior (rostral) process of the palatines, I think the condition on the Ringtail specimen looks different to that on the type specimen of B. darrowi, as Willis (2001) notes. On the Ringtail maxilla the palatine process extends past the rostral extent of the suborbital fenestrae to a point level with the 6th maxillary alveolus; in the B. darrowi holotype it does not extend past the rostral extent of the suborbital fenestrae and is level with the 7th maxillary aveoli. That's pretty different.

While I agree that the premaxilla from Ringtail Site (QM F30319) shares features with the holotype of B. darrowi, and it could be argued that it is assignable to it, I think the maxilla is different, and I agree with the original assignment of Willis (2001) to Baru sp. (for the maxilla). Based on this maxilla and the likely occurrence of material that can be assigned to other crocodylian taxa at this site (e.g, Trilophosuchus and the ‘robust dentary form’), I would be hesitant to conclude that B. darrowi was present in the Ringtail assemblage. I like the idea of the other Baru sp. specimen (a mandibular ramus, QM F31004) possibly pertaining to Trilophosuchus rackhami, but the other specimens mentioned previously are problematic. This causes some issues with regard to the overarching premise of the study: namely that B. darrowi was present in Faunal Zone C. It’s a good idea, but I just don’t think the evidence you have presented is conclusive enough. Similarly, the conclusion that B. wickeni occurred in the Pwerte Marnte Marnte assemblage (and hence outside of Riversleigh) is also very tentative, based purely on the jugal fragment (see previous comments).

A lot of what is discussed in the Discussion may need to be revised in light of some of these issues. I’m ready to be convinced otherwise, but at the moment I not certain that what you are proposing insofar as the temporal and palaeogeographic distribution of B. darrowi and B. wickeni stands up to scrutiny. Finally, I thought it was a bit cheeky to make mention of the NTM P91171-1 skull from Riversleigh 300BR site and bring it into the argument (as an example of how B. wickeni can be used to assign this site to Riversleigh Faunal Zone C). You should probably include this specimen in the Systematic Palaeontology section, particularly if you are going to figure it.

---

## Round 0.2 · Minor Revisions

Dear author,

Again, I must apologise for the lateness of the review (as explained in my email to you). I have accepted the reviewers decision of 'minor revisions'. The suggested changes should be easy to accommodate.

Although your manuscripts acceptance is not contingent on this, the suggestion by reviewer four of splitting your MS into two papers (one dealing with the review/description and another on possible biochronological intrepretations) is something worth considering. Especially as reviewer two remains unconvinced of the biochronological utility of these taxa.

Reviewer 2 ·

Basic reporting

This is a revision of a manuscript I reviewed previously.

I'm now more comfortable that the species referrals are defensible.

The added sections improve the utility of the manuscript, but they appear to have been written rapidly - there are quite a few typos. These should be easily corrected.

I remain unconvinced that these species have any sort of biochronological utility. The biogeographic arguments in the manuscript are more compelling.

Experimental design

no comment

Validity of the findings

see below

Additional comments

p. 4, lone 19 - It is never a good idea to phylogenetically define genera. This is for both philosophical and practical reasons. The philosophical: strictly speaking, genera are not clades; they can, in principle, be paraphyletic. (Otherwise, all living things on earth would belong to the same genus. Either that, or we’d have to accept a system in which genera are nested within other genera.) They’re usually broken apart if found to be paraphyletic, but they’re not conceptually the same as clades. (I actually think one could argue that genera are not really taxa; there are monophyletic groups, and there are species. Where, between the two, are genera? The binomial system is fundamentally incompatible with a strictly phylogenetic taxonomic approach, but we’re stuck with it.) The practical: it can also have unintended consequences – especially if one uses a stem-based definition (as the authors do). Trilophosuchus and Australosuchus are not included as specifiers, so if future analyses draw them closer to B. darrowi than to the other specified taxa, they’d be synonyms of Baru. (I’m not saying such a result is likely, but stranger things have happened.) Or what if we later learn that Mekosuchinae is not monophyletic? What if Baru is closer to Crocodylus than to other Australian Neogene crocodylids? Again, I doubt this would happen - but Baru and Crocodylus would be synonyms should that happen.

p. 4, line 28 – in my view, we should stop using the term “festoon.” It’s meant lots of things over the years; it can refer to concavoconvex patterns in dorsal/ventral view or mediolateral view. Some people use the term for the convexities, others for the concavities. I’m guessing the authors mean convexity in this case; I would just use that word.

p. 5, line 16 – “might also diagnose…” if it’s present in both species, it would presumably diagnose the group including them and not the species themselves. “…might also pertain to…” would be a better way of phrasing this.

p. 9, line 6 – the number of teeth/alveoli can vary by one or two in most modern crocodylian maxillae. Variants are infrequent – the majority of maxillae of a given species will have the same number of alveoli – but they do occur.

p. 9, line 32 – “sub orbital” should be “suborbital”

p. 10, line 3 (and a couple of other places later in the MS) – misspelled “separated”

p. 11, line 6 – only one “forms” is needed.

p. 11, line 22 – “meeth” should presumably be “meets”

p. 11, line 28 – “fossa” should be “fenestra”

p. 12, line 23 – the broad separation of parietal from squamosal is broadly plesiomorphic within Crocodylia; contact is the derived condition where it occurs. (There’s no real reason to specify a figure within the reference here – just cite the paper.)

p. 13, line 5 – space needed between “width” and “of”

p. 13, line 29 – this is only true for basal globidontans; the naris projects dorsally in most modern forms. It also projects anterodorsally in most proximate outgroups to Crocodylia.

p. 14, line 6 – pterygoid height varies ontogenetically. Are the comparisons among specimens of similar size?

p. 14, line 16 – misspelled “aereum” and “pedicel.” I’m also not sure what the authors mean by “pedicel” in this context. (I usually refer to the quadrate ramus, not quadrate pedicel.)

p. 14, line 20 – only one “the” needed before “quadrate,” and misspelled “exposure.”

p. 14, line 28 – extraneous space between “process” and “arises”

p. 15, line 19 – “vagal” isn’t necessarily wrong, but “vagus” is more commonly encountered in the croc literature. The jugular vein passes through here as well.

p. 15, line 23 – misspelled “acutely”

p. 15, line 27 – extraneous “the”

p. 15, line 28 – misspelled “pierced”

p. 15, line 29 – should be ‘…lateral vagal foramen and has a….”

p. 16, line 6 – I suspect “Remarks” should start a new section.

p. 16, line 15 – misspelled “they”

p. 18, line 3 – misspelled “similarly;” “narrower” is a bit less clumsy than “less broad”

p. 19, line 19-20 – might also consider citing Brown et al. (Jour. Anat. 226:322-333, 2015) on variation in the number of premaxillary alveoli. This was noted in several older works, e.g. by Kälin and Brazaitis, though usually only with reference to the number of alveolar positions. Loss of p2 is widespread within Crocodylia; among modern species, Alligator mississippiensis is about the only one where it never happens. (ok - Paleosuchus as well, but it hatches with four premaxillary alveoli. I’ve never seen a Euthecodon premaxilla with more than four alveoli, but I’ve never seen a very small Euthecodon premaxilla.)

p. 19, line 28 – some spaces missing here.

p. 20, line 18 – the Brown et al. reference is relevant here as well.

p. 21, line 18 – misspelled “splits”

p. 21, line 20 – misspelled “Holliday”

p. 25, lines 29-30 – B. wickeni should be italicized.

·

Basic reporting

Being as this MS has already undergone significant review my comments will be limited as the other reviewers have already addressed most of the issues/comments that I had with the MS. I’m glad to see that these issues and suggestions have been addressed and/or amended by the author.

The MS is written in a clear, concise and professional manner. It is well introduced and the aims are clearly stated. Although the material is fragmentary, the data and description of the new crocodylians is thorough and well written, as are the methods. Referencing is relatively extensive.

Experimental design

no comment

Validity of the findings

My biggest concern was that I found the interpretation of the biochronology difficult to follow. Most particularly the linking of the sites - between multiple localities and then within those ‘Faunal Zones’, multiple sites within one location, time interval zones - it was all a bit convoluted and overwhelming trying to understand the whole picture, particularly if one has not worked extensively on the Riversleigh material.
Would a biochronological-strathigraph with Figure 20 (or even also with Figure 21?) help to illustrate this better? Or even possibly, could the MS be split into two: a review/description MS and then a biochrononology/palaeobiogeography interpretation?

Additional comments

Figure 6 & 7: I noticed that you stated in the reply to the reviewers that the image count may be to many – you could combined Figures 6 & 7. This may help the reader as they don’t have to flip between the two.

Figure 21: The position of the ‘key’ for the crocodilians is a bit confusing. Perhaps move it beneath the scale (500km) and/or decrease the size of them.

---

## Round 0.3 · accepted · Accept

Dear author,

I am delighted to inform you that your manuscript has been accepted for publication in PeerJ. Congratulations, and I hope you will use us again as a future publishing venue.